# Co-Seismic Landslides Triggered by the 2014 Mw 6.2 Ludian Earthquake, Yunnan, China: Spatial Distribution, Directional Effect, and Controlling Factors

**Yuying Duan, Jing Luo \***[ID]**, Xiangjun Pei and Zhuo Liu**

State Key Laboratory of Geohazard Prevention and Geoenvironment Protection, Chengdu University of Technology, Chengdu 610059, China; duanyuying@stu.cdut.edu.cn (Y.D.)
\* Correspondence: luoj@cdut.edu.cn; Tel.: +86-02884078689

**Abstract:** The 2014 Mw 6.2 Ludian earthquake exhibited a structurally complex source rupture process and an unusual spatial distribution pattern of co-seismic landslides. In this study, we constructed a spatial database consisting of 1470 co-seismic landslides, each exceeding 500 m$^2$. These landslides covered a total area of 8.43 km$^2$ and were identified through a comprehensive interpretation of high-resolution satellite images taken before and after the earthquake. It is noteworthy that the co-seismic landslides do not exhibit a linear concentration along the seismogenic fault; instead, they predominantly extend along major river systems with an NE–SW trend. Moreover, the southwest-facing slopes have the highest landslide area ratio of 1.41. To evaluate the susceptibility of the Ludian earthquake-triggered landslides, we performed a random forest model that considered topographic factors (elevation, slope, aspect, distance to rivers), geological factors (lithology), and seismic factors (ground motion parameters, epicentral distance, distance to the seismogenic fault). Our analysis revealed that the distance to rivers and elevation were the primary factors influencing the spatial distribution of the Ludian earthquake-triggered landslides. When we considered the directional variation in ground motion parameters, the AUC of the model slightly decreased. However, incorporating this variation led to a significant reduction in the proportion of areas classified as "high" and "very high" landslide susceptibility. Moreover, SED$^d$ emerged as the most effective ground motion parameter for interpreting the distribution of the co-seismic landslides when compared to PGA$^d$, PGV$^d$, and Ia$^d$.

**Keywords:** 2014 Mw 6.2 Ludian earthquake; co-seismic landslide susceptibility; random forest; ground motion parameters; directional effect





## 1. Introduction

Earthquakes occurring in hilly regions can trigger extensively destructive landslides, which has aroused widespread concerns worldwide [1–6]. Notably, the damage caused by co-seismic landslides may sometimes exceed that resulting from violent shaking and fault ruptures [7–12]. Over the past decades, numerous studies have been conducted on co-seismic landslides in mountainous areas, such as the 2004 Mw 6.6 mid-Niigata earthquake [13], the 2005 Mw 7.6 Northern Pakistan earthquake [14], the 2008 Ms 8.0 Wenchuan earthquake [15], the 2018 Mw 6.6 Hokkaido Eastern Earthquake [16], and the 2020 Mw 6.9 Samos earthquake [17].

The distribution of co-seismic landslides is influenced by various factors, such as topography, hydrology, lithology, seismic factors, and human activity, etc. [1,18]. Numerous studies have shown that the magnitude of an earthquake influences the probability of landslide occurrence (e.g., Keefer [1,19]; Rodriguez et al. [20]). Therefore, seismic factors, including ground motion parameters, epicentral distance, and distance to the seismogenic fault are closely related to the spatial distribution of co-seismic landslides. Through statistical analysis of historical co-seismic landslides, some scholars have proposed that, in

determining the extent of co-seismic landslides, the distance from the surface projection of the seismogenic fault is a more crucial factor compared to epicentral distance [21–23].

In addition, the correlation between ground motion parameters (i.e., Peak Ground Acceleration (PGA); Peak Ground Velocity (PGV); Arias Density (Ia)) and the occurrence of co-seismic landslides has been widely investigated [24–27]. Several studies have indicated that sufficient energy is a prerequisite for landslide occurrence. Hence, Ia, which incorporates the amplitude, spectrum, and total duration characteristics of ground motion, can more comprehensively reflect earthquake information [28]. As a result, it is considered a reliable parameter for predicting co-seismic landslides compared to relying solely on ground shaking amplitudes [29–34]. Similarly, the Specific Energy Density (SED), which integrates velocity over the duration of an earthquake, has been pervasively used as a ground motion parameter to quantify the intensity of seismic energy [35–38]. Moreover, there is a positive correlation between SED and the intensity of earthquake damage [39,40]. Unfortunately, there is a lack of studies that specifically investigate the relationship between the SED and the occurrence of co-seismic landslides.

It should be noted that the significant "directional effect" of the spatial distribution of co-seismic landslides has been frequently reported after many large earthquakes, including the 1999 Mw 7.6 Taiwan Chi-Chi earthquake, the 2005 Mw 7.6 Pakistani Kashmir earthquake [41], and the 2008 Mw 8.0 Wenchuan earthquake [42]. This effect implies that slopes facing towards the earthquake source are less susceptible to landslides compared to slopes aligned in the same dip direction as the propagation of seismic waves. Some researchers attribute this phenomenon to the fact that seismic waves exhibit varying responses on slopes in different directions, resulting in different degrees of instability [43–45]. Chen et al. [46] provided insight into the "directional effect" of the co-seismic landslides in the 2018 Mw 6.6 Hokkaido earthquake using Discontinuous Deformation Analysis (DDA) and demonstrated that the difference of seismic energy in different slope aspects determines whether a landslide will occur or not. This finding is consistent with the spatial distribution of co-seismic landslides and seismic wave characteristics observed in this earthquake. In addition, the researchers discovered that the preferred collapse direction of houses during earthquakes is highly related to the direction of the highest seismic energy [47–50].

On 3 August 2014, a moment magnitude of 6.2 (Ms 6.5) earthquake struck Ludian County in Yunnan Province, southwestern China, at 08:30:11 UTC (16:30:10.2 Beijing Standard Time). The earthquake's epicenter was located at coordinates 27.11°N and 103.35°E [51,52]. It had a shallow focal depth of 12 km and was characterized by a complex faulting process involving Riedel shear structures [53,54], resulting in a massive surface rupture zone [55] and a large number of the co-seismic landslides [2]. The largest landslide triggered by the Ludian earthquake was the Hongshiyan landslide, with a volume of approximately 12.24 Mm$^3$. Interestingly, the opposite slope, which has a similar geologic structure but steeper topography, does not show obvious deformation associated with the earthquake [56]. By utilizing the high-quality seismic metadata from 28 strong motion stations provided by the China Earthquake Administration, we were able to obtain measurements of PGA, PGV, Ia, and SED in four cardinal directions (i.e., east, west, south, and north). It was found that the values of the aforementioned parameters in the direction facing the Hongshiyan slope are significantly larger than those in the opposite slope. Zou et al. [57] performed a spatial analysis using GIS to investigate and characterize the correlations between the landslide occurrence and various influencing factors in the 2014 Ludian earthquake. Their results suggested that two factors, namely distance to the seismogenic fault and slope gradient, are the most critical in determining the spatial distribution of co-seismic landslides. However, their study utilized intensity as the ground motion parameter instead of any of commonly used parameters such as PGA, PGV, Ia, or SED. Chen et al. [58] used only the critical acceleration in the Newmark method to assess the susceptibility of co-seismic landslides triggered by the Ludian earthquake due to the unavailability of seismic record. The critical acceleration took into account various geological factors, including terrain, lithology, physical properties of rock and soil, and

other relevant factors. On the other hand, statistical models [59,60], which encompass support vector machine (SVM) [61,62], logic regression (LR) [63,64], and random forest (RF) [65,66], are currently widely used methodologies for co-seismic landslide hazard assessment. The mathematical relationships between landslides and their triggering factors are derived using co-seismic landslide inventories in these models. An overview of the state-of-the-art techniques for earthquake-induced landslides susceptibility assessment is presented by Shao et al. [67]. In this study, we constructed a spatial database of the 2014 Ludian earthquake-triggered landslides. Then, we used a random forest model to conduct co-seismic landslide hazard assessment, where SED was first considered as the ground motion parameter. The primary objectives of the study are as follows: (1) to elucidate the spatial distribution pattern of co-seismic landslides in the earthquake, an earthquake caused by a complex faulting process; (2) to determine the ground motion parameter that most significantly contributes to co-seismic landslides in the earthquake among PGA, PGV, Ia, and SED; and (3) to examine the relationship between seismic energy variations in different directions and the corresponding varying degrees of slope instability.

## 2. Study Area and Materials

### 2.1. Study Area

The 2014 Ludian earthquake occurred in the western segment of the Zhaotong–Lianfeng fault zone. The study area is located to the east of the Xianshuihe–Xiaojiang fault system, which lies between the Sichuan–Yunnan block, southeastern Tibetan Plateau. Previous investigations indicated that this region is characterized by numerous faulting activities and a frequent occurrence of large earthquakes [68]. The primary active faults in the study area are the NW-trending Baogunao–Xiaohe fault and the NE-trending Zhaotong–Ludian fault, as shown in Figure 1. The Baogunao–Xiaohe fault was the seismogenic fault of the Ludian earthquake and is characterized by a complex faulting process associated with Riedel shear structures [53,54].

The study area features a "V-shaped" high mountain valley landscape with an average stream slope of 1.22% and a natural drop of approximately 220 m. The exposed strata in the area range from the Sinian System to Quaternary System, except for the Cretaceous System. In particular, the Paleozoic Erathem is the most widespread stratum in the area, with the Permian System having the largest area of exposure, followed by the Cambrian System. The epicenter of the 2014 Ludian earthquake is within the Permian stratum. The lithology of the Paleozoic Erathem is primarily composed of sandstone, shale, limestone, and dolomite. Substantial joints and cracks develop in these strata and are most commonly observed in the northeast and southeast regions of the study area. The Mesozoic System in the study area is characterized by continental detrital deposits, with sandstone and mudstone being the predominant lithological composition. In contrast, the Cenozoic System is scattered in intermountain basins, such as the Ludian basin, and is primarily composed of fluvio-lacustrine facies of clayey rock and sandy gravel beds. Intensive crustal deformation has resulted in strong faulting and rock fracturing in the region, with weathering particularly evident along faults and crush belts [69,70].

A total of 115 earthquakes with magnitudes equal to or greater than 4.7, occurring between 624 CE and 2014 CE within a 200 km radius around the epicenter of the 2014 Ludian main shock, were extracted by the China Earthquake Administration. It is evident that while the Ludian earthquake is classified as a moderate event, it is also considered a low-frequency seismic occurrence in this region, with an estimated recurrence interval of about 100 years [52].

The main watercourse in the study area is the Niulan River, which flows predominantly from southeast to northwest, carving steep valleys and penetrating the mountains to depths ranging from 1200 to 3300 m. Meanwhile, the Shaba and Longquan rivers run from northeast to southwest, creating numerous terraces at various elevations. The altitude of the study area varies from 500 to 3900 m. Approximately 60% of the slopes have gradients between 10° and 30°, and those steeper than 40° are mainly distributed along the Niulan

River. The terrain in the northern and eastern regions is predominantly flat, with low mountains and hills characterized by slopes generally less than 20°.

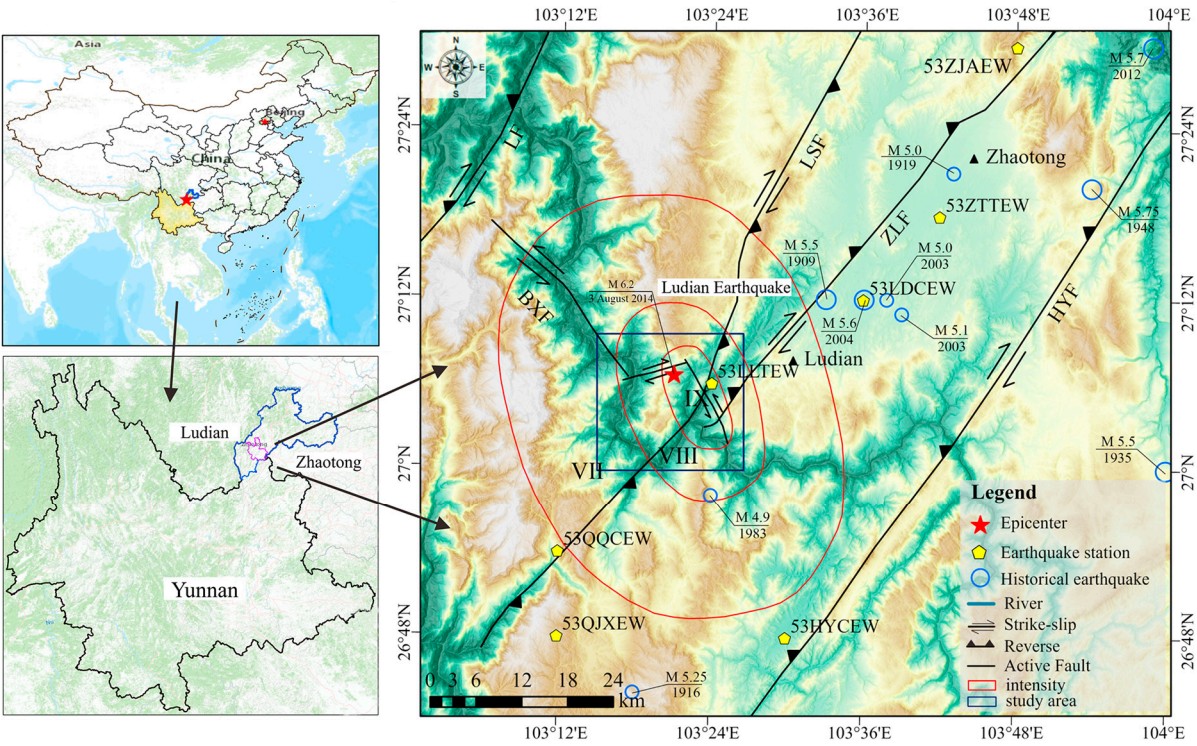

**Figure 1.** Maps showing the geological setting of the study area and the location of the 2014 Ludian earthquake. The following faults are indicated: HYF: Huize–Yanjin fault; ZLF: Zhaotong–Ludian fault; LSF: Longshu fault; LF: Lianfeng fault; BXF: Baogunao–Xiaohe fault. Intensity map published by the China Earthquake Administration. Source: http://www.gov.cn/xinwen/2014-08/07/content_2731360.html, accessed on 1 October 2022. The fault map was derived from Luo et al. [52,53].

### 2.2. Landslide Inventory

Landslide inventories play a crucial role in analyzing the spatial distribution of landslides, evaluating their causative failure mechanisms, and generating susceptibility maps. The interpretation of the 2014 Ludian earthquake-triggered landslides was conducted by using an extensive collection of high-resolution satellite images captured before and after the earthquake. These images comprised Sentinel-2A images (10 m resolution), GF-1 images (2 m resolution), GF-2 images (1 m resolution), Google Earth data (0.5 m resolution), and UAV (0.2 m resolution) (Table 1). All images were subjected to geometric correction, enhancement, and coordinate system conversion. In order to address the potential overestimation of total landslide volumes that can occur with automated methods when nearby individual events are inadvertently combined into a single entity, we opted for manual digitization over automated extraction techniques to identify landslides in this study. The boundaries of landslides were manually digitalized into polygons using satellite images within a GIS environment. To guarantee the accuracy of the landslide inventory derived from remote sensing interpretation, we conducted an extensive field investigation within the densely populated area affected by co-seismic landslides, lasting for approximately 30 days. Based on the analysis of satellite images and the findings from the field investigations, we identified a total of 1470 landslides larger than 500 m$^2$ within a rectangular area of approximately 360 km$^2$. The total area covered by these landslides is estimated to be around 8.43 km$^2$. The landslide number density (LND) obtained in this study is 1470/360 km$^2$ = 4.083/km$^2$ and the landslide area percentage (LAP) is (8.43 km$^2$/360 km$^2$) × 100% = 2.34%.

**Table 1.** Information of data used.

| No. | Data Type | Date | Resolution (m) |
| --- | --- | --- | --- |
| 1 | Sentinel-2A | 1 August 2014 | 10 |
| 2 | GF-1 | 26 October 2014 | 2 |
| 3 | GF-2 | 14 February 2015 | 1 |
| 4 | Google Earth data | 30 January 2014, 20 August 2014 | 0.5 |
| 5 | UAV | 15 September 2015 | 0.2 |

After the 2014 Ludian earthquake, some researchers interpreted the co-seismic landslides. The most comprehensive interpretation of landslides to date was performed by Wu et al. [71] in 2020, which yielded a landslide number of 12,817. They also compared these with other results in terms of the quality and resolution of remote sensing images, coverage, interpretation method, landslide number, and landslide area, indicating that their results are more complete, detailed and objective. The coverage area interpreted in their study was larger than that of our study. Additionally, in our study, all the interpreted landslide targets had an area greater than 500 m$^2$. Despite these differences, it is noteworthy that the LAP obtained in our study (2.34%) is relatively close to the result of their study (2.71%). This similarity suggests that the landslide database we established possesses a certain level of representativeness and can be considered reliable for assessing co-seismic landslide susceptibility.

Since the primary focus of landslide susceptibility analysis revolves around the identification of potential landslide source areas, only the source areas can be used to analyze the spatial distribution pattern of landslides and to train the susceptibility model [72]. Therefore, on the basis of the above co-seismic landslide database, a landslide source area database was established based on Google Earth satellite images and UAV images to differentiate between the source area, circulation area, and accumulation area of the landslide. Specifically, in cases where it was challenging to clearly identify the provenance area of a landslide, we selected the upper 50% of the landslide based on the elevation values and designated it as the provenance area [73]. Subsequently, mass points corresponding to the identified provenance areas were then extracted and employed as landslide sample points for further analysis. In addition, it is crucial to emphasize that all the landslide areas mentioned below specifically refer to the landslide source areas.

To ensure a balanced model, an equal number of negative samples were generated in our study. There are various methods available for generating negative samples. Selecting negative samples from areas with lower landslide occurrence probabilities is a valuable approach that can greatly enhance the reliability of landslide susceptibility prediction [16,18]. Two commonly used methods for negative sample selection are random sampling [74,75] and buffer-controlled sampling [76,77]. However, these methods have a limitation in that they cannot guarantee the selection of non-landslide samples from areas with extremely low susceptibility levels. To overcome this limitation, the Information Value model [78,79] and the Mean Clustering model [80] have been preliminarily applied to study of landslide negative sample selection. The Fuzzy c-means algorithm (FCM) possesses the capability to classify the study area into different levels of landslide susceptibility based on the geographical attributes of landslide controlling factors. This categorization allows for the generation of non-landslide samples from areas with low susceptibility levels. Notably, FCM is not influenced by subjective factors and operates independent of any specific model. In a comparative analysis conducted by Liang [80] on selecting non-landslide samples for assessing shallow landslide susceptibility using machine learning, FCM outperformed the K-means algorithm in terms of sampling reliability. Consequently, this study chose to utilize the FCM for the extraction of non-landslide samples. We employed FCM to categorize landslide susceptibility in the study area into five classes, i.e., "Very low", "Low", "Moderate", "High", and "Very high" (Figure 2). The "Very low" and "Low" areas were selected as non-landslide sampling areas, from which a total of 1470 negative samples were

randomly generated. This approach ensures a representative set of non-landslide samples for our analysis.

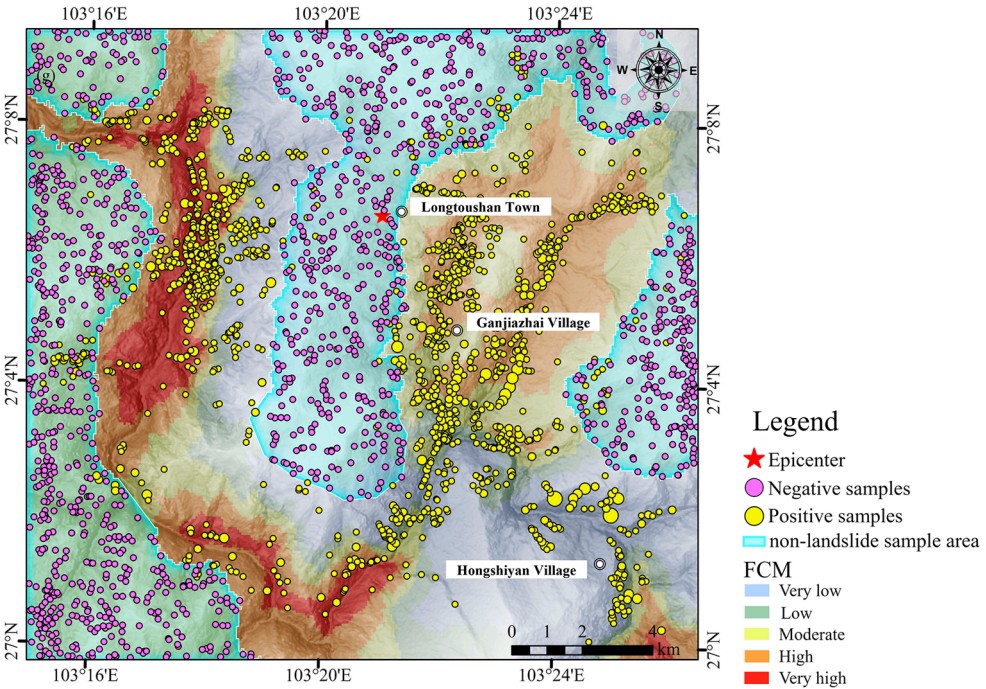

**Figure 2.** Landslide and non-landslide sampling areas.

### 2.3. Spatial Distribution Pattern

#### 2.3.1. Landslide Number Density

Figure 3 shows the graphical representation of the number density of landslides triggered by the 2014 Ludian earthquake. The results reveal that the highest LND, landslides/km$^2$), which is up to 50, is primarily concentrated in the region southeast and west of the epicenter, specifically at a distance of 5–9 km. The northern zone of the epicenter exhibits a lower LND compared to the southern zone. Notably, these co-seismic landslides are not distributed linearly along the seismogenic fault nor clustered around the epicenter region. Instead, the landslides primarily spread along the Niulan, Shaba, and Longquan rivers. When investigating the underlying formation mechanism behind this specific distribution pattern, it is essential to consider the following factors. Firstly, almost 60% of the slopes in the study area range from 10° to 30°, while slopes steeper than 40° are mainly distributed along the Niulan, Shaba, and Longquan rivers. The topographic amplification of seismic responses significantly increases the susceptibility of these steeper slopes to instability. Secondly, due to the process of valley incision, the slopes along the riverbanks exhibit steep terrain and undergo substantial weathering and unloading. Consequently, shallow landslides, which represent the primary type of landslides induced by the Ludian earthquake [58], are highly prone to occur. Lastly, the proximity of the rivers to the epicenter and seismogenic fault implies that the intense shaking also affects the slopes along the riverbanks. However, in areas closer to the epicenter and seismogenic fault, the landscape is predominantly characterized by gentle slopes. This suggests that more intense shaking is required to initiate sliding and that there is relative stability under dynamic seismic conditions. This finding coincides with the results revealed by Chen et al. [81].

#### 2.3.2. Landslide Area Percentage

In addition to LND, landslide size is another critical parameter for investigating the distribution characteristics of co-seismic landslides. This is because the size of a landslide can determine the extent of damage and the type of mitigation measures required. Based on the classification of landslide source areas presented in Figure 4, it was found that 67%

of the large landslides were primarily concentrated in the region southeast of the epicenter. The provenance area of the largest landslide is approximately 0.2 km$^2$.

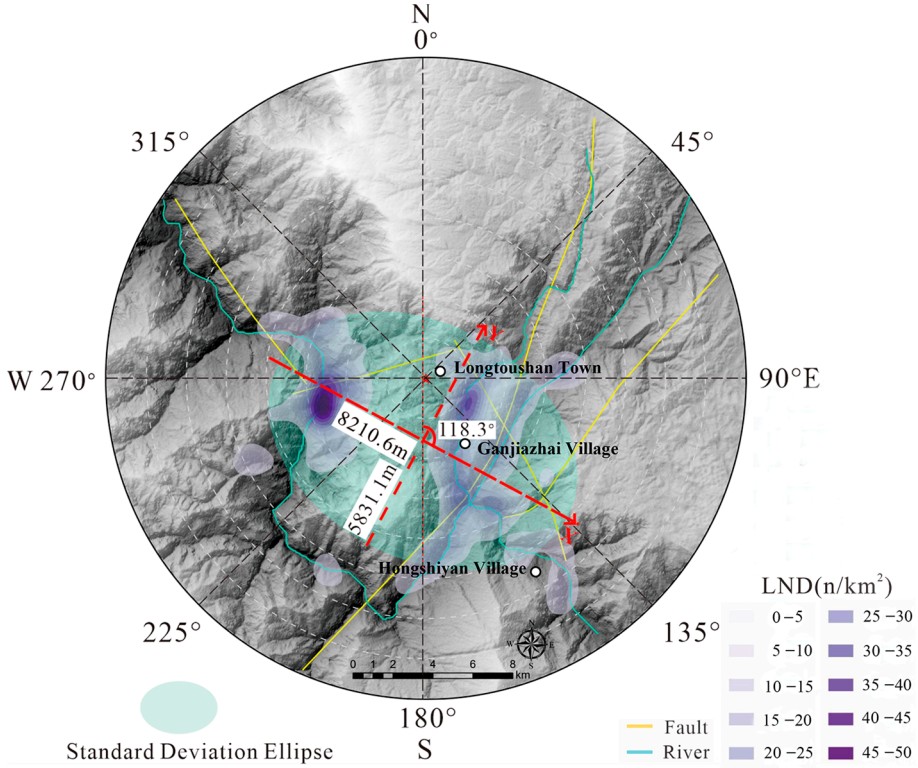

**Figure 3.** Density distribution of co-seismic landslides in the Ludian earthquake. The ellipses are the standard deviation ellipse plotted with the number of landslides as weight.

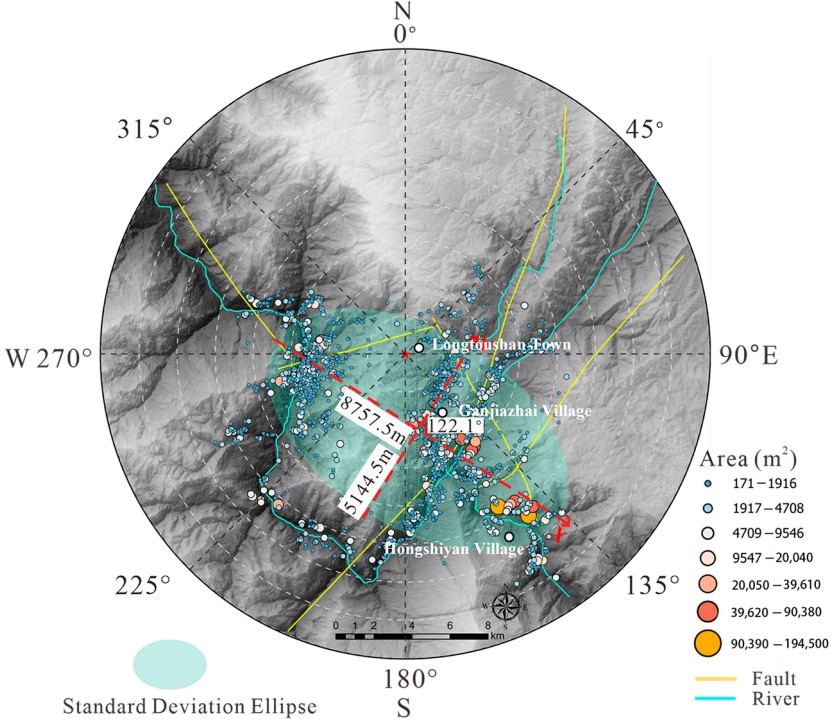

**Figure 4.** Area distribution of co-seismic landslides in the Ludian earthquake. The ellipses are the standard deviation ellipse plotted with the landslide source area as weight.

Figure 5 displays the spatial distribution of co-seismic landslides in the study area, with landslide-affected areas divided into squares of 0.1 km². Each square provides comprehensive information on the LND and LAP.

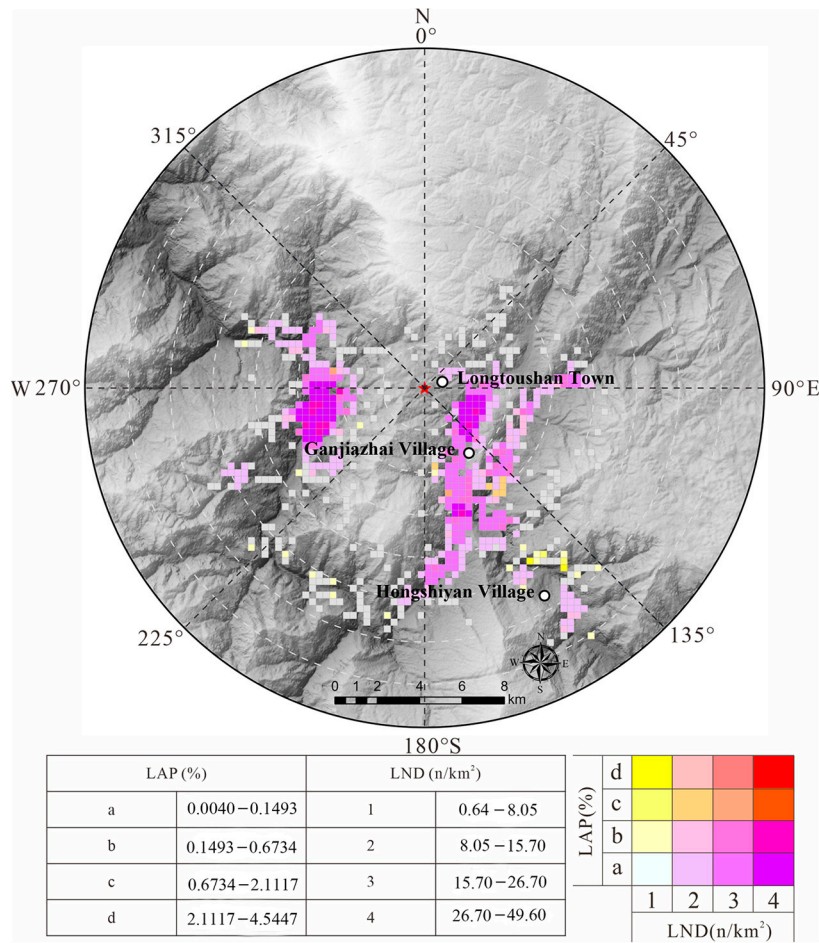

**Figure 5.** The LAP–LND distribution of the Ludian co-seismic landslides. The provenance areas of landslides are divided into square grids of 0.1 km² each.

The parameters LND and LAP were effectively connected by employing a color scale that integrated the two individual color scales, as shown in Figure 5. The color scale combined four classes for each parameter, with LND ranging from 1 to 4 (increasing number density) and LAP ranging from "a" to "d" (increasing area percentage). This resulted in a total of 16 classes, ranging from 1a (low number density of small landslides) to 4d (high number density of large landslides). The size of the classes for both landslide density and dimension distribution was carefully selected using the Jenks natural breaks classification method to define the optimal value distribution between the classes [82].

The regional classification of co-seismic landslides reveals that their distribution is primarily characterized by landslides with high density but relatively small dimensions. This feature is remarkably obvious in the regions west and east to the epicenter. Large landslides with low density are mainly concentrated in the southeastern river valley, which is the farthest from the epicenter. There are no high density and large dimension landslides distributed.

### 2.3.3. Standard Deviational Ellipse

The Standard Deviational Ellipse (SDE) of LND and LAP was used to quantify the spatial pattern of co-seismic landslides, as shown in Figures 3 and 4. The results indicate that both densities have an NE–SW trend (Table 2). Furthermore, the ellipse center of LAP

is shifted to the southeast, suggesting that this region encompasses the highest number of large landslides. The lengthening of the X-axis and shortening of the Y-axis, together with the reduction in elliptical flatness and the increase in centripetal force, indicate a distinct directional trend of LAP. The increase in the spatial rotation angle and the rotation of the ellipse direction from SEE to SSE suggest that landslides are more concentrated in the southeast. Additionally, the decrease in elliptical area demonstrates that the spatial dispersion of landslide source area was reduced and became more concentrated.

**Table 2.** Standard Deviational Ellipse of the spatial distribution of LND and LAP.

|  | CenterX | CenterY | XStdDist (m) | YstdDist (m) | Angle | Area (km$^2$) |
|---|---|---|---|---|---|---|
| SDE (LND) | 103.349 | 27.082 | 8210.6 | 5861.1 | 118.3 | 150.4 |
| SDE (LAP) | 103.358 | 27.074 | 8757.5 | 5144.5 | 122.1 | 140.8 |

CenterX: longitude of the ellipse center; CenterY: latitude of the ellipse center; XStdDist: length of the X-axis; YstdDist: length of the Y-axis; Angle: direction angle of the ellipse; Area: the area of the ellipse.

### 2.3.4. Slope Aspect

Chigira et al. [83] demonstrated that the slope aspect (slope facing direction) has a strong control on the distribution of co-seismic landslides, and some slope aspects are more susceptible to co-seismic landslides than others. To accurately assess the susceptibility of co-seismic landslides to different slope aspects in the study area, we used the landslide area ratio proposed by Chen et al. [81] to eliminate the influence of the area ratio of slope aspect. The slope aspect was divided into eight data sections based on different azimuths: N (337.5, 360.0) and (0, 22.5°), NE (22.5, 67.5°), E (67.5, 112.5°), SE (112.5, 157.5°), S (157.5, 202.5°), SW (202.5, 247.5°), W (247.5, 292.5°), and NW (292.5, 337.5°). Based on this, we calculated the landslide area ratio of each section, excluding the areas with flat terrain (Table 3 and Figure 6).

**Table 3.** Landslide area ratios for different data sections of slope aspect.

| Aspect | $A_i$ (km$^2$) | $DA_i$ (km$^2$) | $A_i/A$ | $DA_i/DA$ | $R_i$ |
|---|---|---|---|---|---|
| N | 40.557813 | 0.368961 | 0.111869 | 0.086252 | 0.771013 |
| NE | 38.280625 | 0.237103 | 0.105588 | 0.055428 | 0.524946 |
| E | 41.409688 | 0.509088 | 0.114218 | 0.119010 | 1.041950 |
| SE | 53.787188 | 0.825668 | 0.148359 | 0.193017 | 1.301016 |
| S | 43.641719 | 0.631193 | 0.120375 | 0.147555 | 1.225792 |
| SW | 43.391094 | 0.721878 | 0.119684 | 0.168754 | 1.410000 |
| W | 50.311563 | 0.570736 | 0.138772 | 0.133421 | 0.961443 |
| NW | 51.168438 | 0.413066 | 0.141136 | 0.096563 | 0.684184 |

$A_i$: the area of portion of data section $i$ with the slope aspect; $DA_i$: the landslide source area with the data section $i$ of the slope aspect; $A$: the total area of the study area (360 km$^2$); $DA$: the total landslide source-occupying area in the study area (4.28 km$^2$); $R_i$: the local landslide area ratio for the data section $i$ of a slope aspect.

The study area exhibits a generally uniform slope aspect distribution, with the slopes facing SE, NW, and W accounting for approximately 15% of the total area, while the remaining slope aspects account for approximately 11%. The $R_i$ values for slopes facing SE, S, and SW are 1.30, 1.23, and 1.41, respectively. In contrast, the $R_i$ values for other slope aspects are less than or close to 1. Particularly, the $R_i$ values for N-, NW-, and NE-oriented slopes are as low as 0.77, 0.68, and 0.52, respectively, indicating a low susceptibility to landslides in these areas. In total, the percentage of landslide source area on slopes facing south is much larger than that on slopes facing north, indicating a higher susceptibility to co-seismic landslides on slopes facing SW, SE, and S.

The distribution pattern of LND, LAP, and landslide slope aspect indicate that the Ludian earthquake-triggered landslides have obvious "directional effects", both in terms of their locations and the corresponding slope aspects. Therefore, the directional difference of seismological signals should be considered when evaluating the susceptibility of co-seismic landslides.

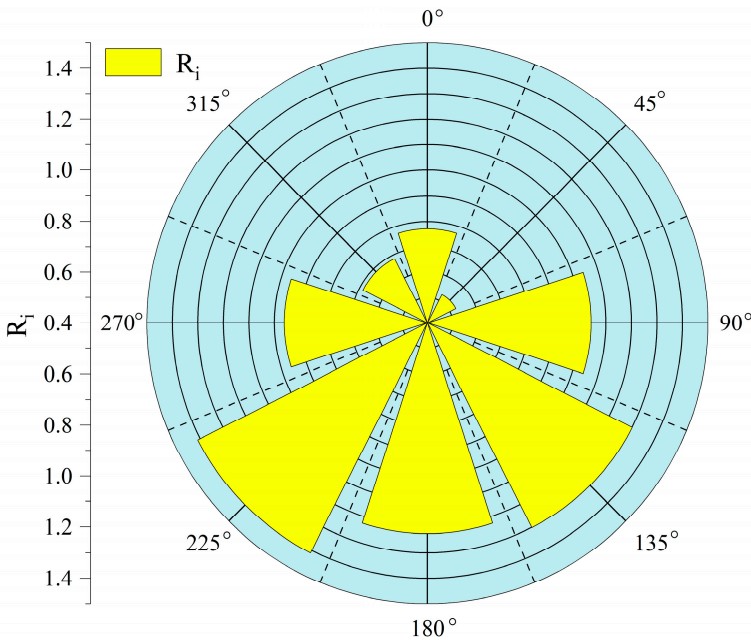

**Figure 6.** Relation of co-seismic landslide area abundances with slope aspect: 0° (North), 90° (East), 180° (South), 270° (West).

## 3. Methodology

### 3.1. Selection and Analysis of Landslide Controlling Factors

#### 3.1.1. Selection of Landslide Controlling Factors

The reasonable selection of controlling factors significantly impacts the accuracy and authenticity of landslide susceptibility assessment results. However, there is currently no unified standard for choosing controlling factors. Ayalew et al. [63] suggested that the criteria for selecting controlling factors should be based on their similarity to landslide locations, measurability, non-redundancy, and knowledge of the geo-environmental conditions of the study area. Therefore, taking guidance from the previous research on the co-seismic landslide susceptibility assessment [43,67,72,73], we carefully identified and selected 10 factors that are strongly associated with the occurrence of co-seismic landslides, considering three key perspectives: seismic, topographic, and geological aspects (Tables 4 and 5). Remarkably, in this study, SED was introduced as one of the ground motion parameters to assess co-seismic landslide susceptibility at the regional scale unprecedentedly. The digital elevation model (DEM) utilized in this study was acquired from the Advanced Land Observing Satellite (ALOS) with a spatial resolution of 12.5 m. From the DEM, we derived raster maps of aspect, slope, Topographic Roughness Index (TRI), and Topographic Wetness Index (TWI), which provided crucial information into the topographic characteristics of the study area. Meanwhile, raster maps of epicentral distance, distance to the seismogenic fault, and distance to rivers were generated with a 1 km buffer unit. The Lithology data used in the analysis were obtained from the China Geological Survey (Table 4). The ground motion parameters, including PGA, PGV, Ia, and SED, were derived from the high-quality metadata from 28 strong motion seismic stations provided by the China Earthquake Administration. To generate raster maps of the ground motion parameters, Kriging interpolation was employed.

SED is defined as the squared velocity at any given time, integrated over the entire time range. The definition of SED is as follows:

$$\text{SED} = \int_0^T [v(t)]^2 dt \tag{1}$$

where $v$ is the velocity, $T$ is the entire time.

**Table 4.** Descriptions of categorized lithology in the study area.

| No. | Stratum | Lithology Description |
|-----|---------|----------------------|
| 1 | D1 | Lower Devonian System. Clastic rocks |
| 2 | D2 | Middle Devonian System. Quartz sandstone, siltstone, dolomite |
| 3 | O1 | Lower Ordovician System. Fine sandstone, dolomite, mica siltstone |
| 4 | O2 | Middle Ordovician System. Dolomite, sandstone with shale and argillaceous limestone |
| 5 | O3 | Upper Ordovician System. Dolomite, sandstone with shale and argillaceous limestone |
| 6 | P1 | Lower Permian System. Siltstone, shale, limestone |
| 7 | P2 | Upper Permian System. Mudstone, porphyritic basalt, volcanic breccia |
| 8 | S2 | Middle Silurian System. Shale, carbonatite, clastic rocks |
| 9 | T1 | Lower Triassic System. Siltstone, argillaceous siltstone with fine sandstone |
| 10 | Z1 | Lower Sinian System. Basal conglomerate, pebbly sandstone, sandstone, quartz sandstone |
| 11 | Z2 | Upper Sinian System. Dolomite, dolomite limestone, dolomitic shale |
| 12 | Є1 | Lower Cambrian System. Sandstone, shale, dolomite, argillaceous limestone |
| 13 | Є2 | Middle Cambrian System. Gray dolomite, shale with siltstone, clastic rock, argillaceous limestone |
| 14 | Є3 | Upper Cambrian System. Gray dolomite, shale with siltstone, clastic rock, argillaceous limestone |

**Table 5.** Information on landslide controlling factors.

| Factor | Variable | Data Source | Resampled Resolution |
|--------|----------|-------------|----------------------|
| | PGA | | 12.5 m |
| | PGV | | " |
| | SED | | " |
| | Ia | | " |
| Seismic factor | PGA$^d$ | China Earthquake Administration | " |
| | PGV$^d$ | | " |
| | SED$^d$ | | " |
| | Ia$^d$ | | " |
| | ED | | " |
| | DSF | | " |
| | Elevation | | " |
| | Aspect | | " |
| Topographic factor | Slope | ALOS DEM | " |
| | TRI | | " |
| | TWI | | " |
| | DR | | " |
| Geological factor | Lithology | China Geological Survey | " |

$''$: It is stated that the resolution of all the data below is the same as the resolution of the first row, which is 12.5 meters.

Ia represents the integral of the squared acceleration history, which is expressed as follows:

$$\mathrm{Ia} = \frac{\pi}{2g} \int_{t_1}^{t_2} [a(t)]^2 dt \tag{2}$$

where $a$ is acceleration history (m/s$^2$), $t_1$ and $t_2$ define the total duration of the acceleration history, and $g$ is the gravity (m/s$^2$).

The four cardinal directions were defined as follows: $(0, 45°)$ and $(315, 360°)$ as north, $(45, 135°)$ as east, $(135, 225°)$ as south, and $(225, 315°)$ as west. The different values of PGA, PGV, SED, and Ia in each of these four cardinal directions were extracted separately. Subsequently, the ground motion factor maps considering the directional effects were combined using the raster mosaic tool in ArcGIS. This process generated raster maps of direction-dependent PGA, PGV, SED, and Ia, which were abbreviated as PGA$^d$, PGV$^d$, SED$^d$, and Ia$^d$, respectively (Table 5). The raster maps of PGA, PGV, SED, and Ia, as well as their corresponding direction-dependent factors are shown in Figure 7, while those of other factors are presented in Figure 8. It should be noted that topographic roughness

index, topographic wetness index, epicentral distance, distance to the seismogenic fault, and distance to rivers are abbreviated as TRI, TWI, ED, DSF, and DR for convenience in the following figures, tables, and context.

### 3.1.2. Correlation Analysis of Landslides and Controlling Factors

To evaluate the potential contribution of individual controlling factors in predicting landslide occurrences, a correlation analysis was conducted between the controlling factors and the incidence of landslides. The relationship between the continuous factors (i.e., elevation, aspect, slope, TRI, TWI, ED, DSF, DR, PGA, PGV, SED, Ia, PGA$^d$, PGV$^d$, SED$^d$, and Ia$^d$) and landslide occurrences was quantified using the point biserial correlation coefficient ($r_{pb}$) [84]. The correlation coefficient yields a value within the range of $-1$ to 1, where 0 denotes no correlation, $-1$ represents perfect anticorrelation, and 1 indicates perfect correlation. On the other hand, Cramer's V ($V$) was employed to analyze the relationship between nominal factors (lithology) and landslide occurrences [85]. Cramer's V produces a value ranging from 0 to 1, with higher values approaching 1 indicating a stronger relationship.

$$r_{pb} = \frac{M_1 - M_0}{S_N} \sqrt{\frac{n_1 n_0}{n^2}} \tag{3}$$

where $M_1$ and $M_0$ represent the means of the two sets of data, respectively, $n_1$ and $n_0$ represent the number of data in each set, $n$ represents the total number of data, and $S_N$ represents the pooled standard deviation of all data.

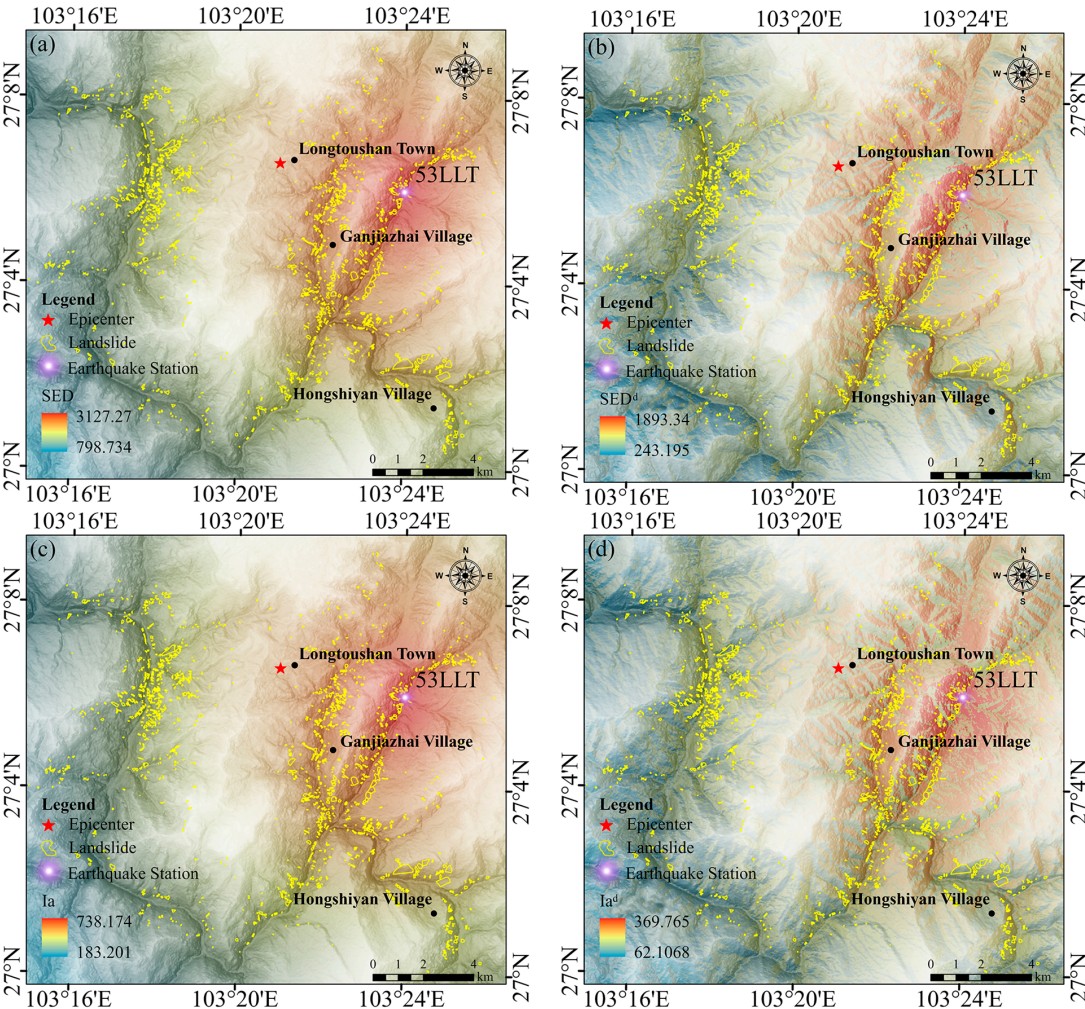

**Figure 7.** *Cont.*

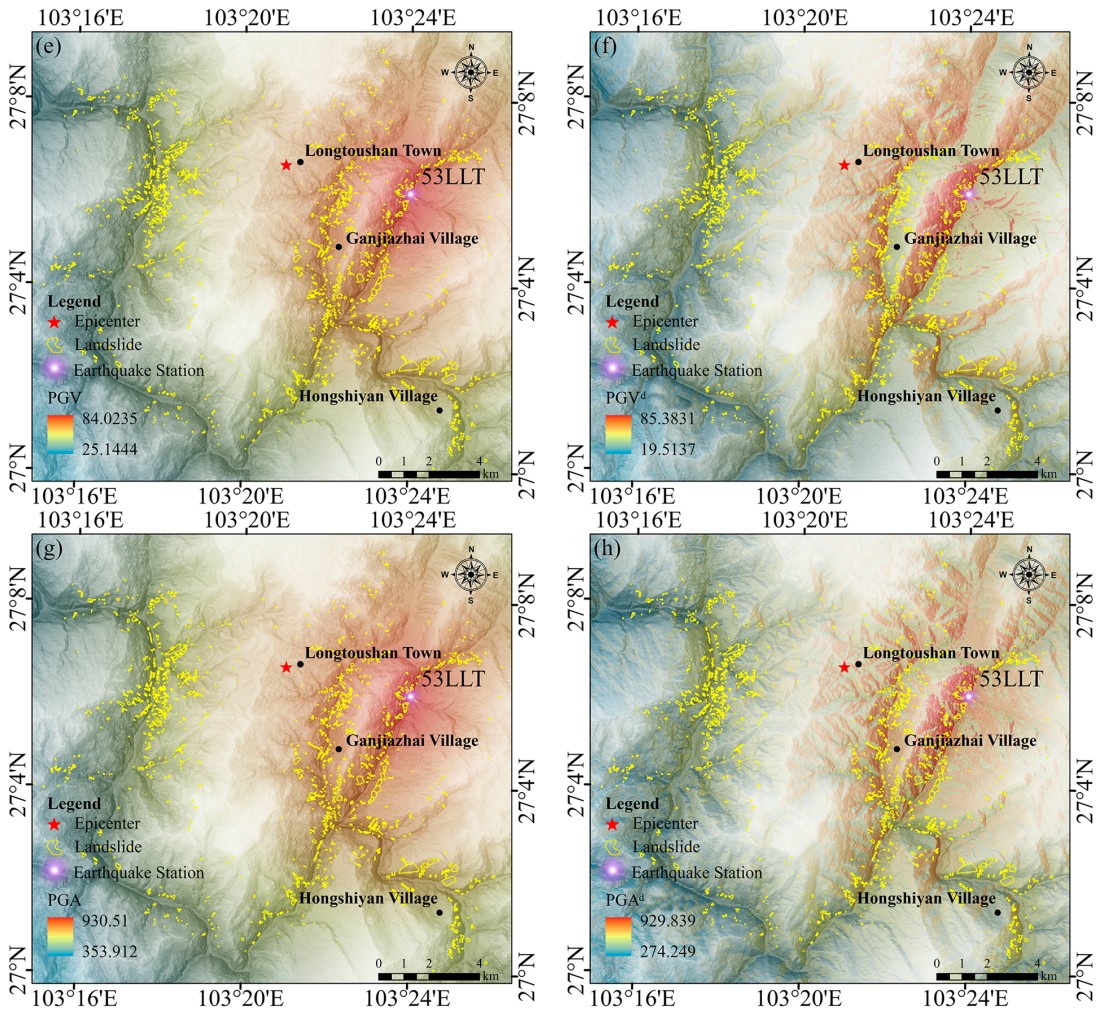

**Figure 7.** Raster maps of SED, Ia, PGV, PGA, (**a**,**c**,**e**,**g**) and SED[d], Ia[d], PGV[d], PGA[d] (**b**,**d**,**f**,**h**).

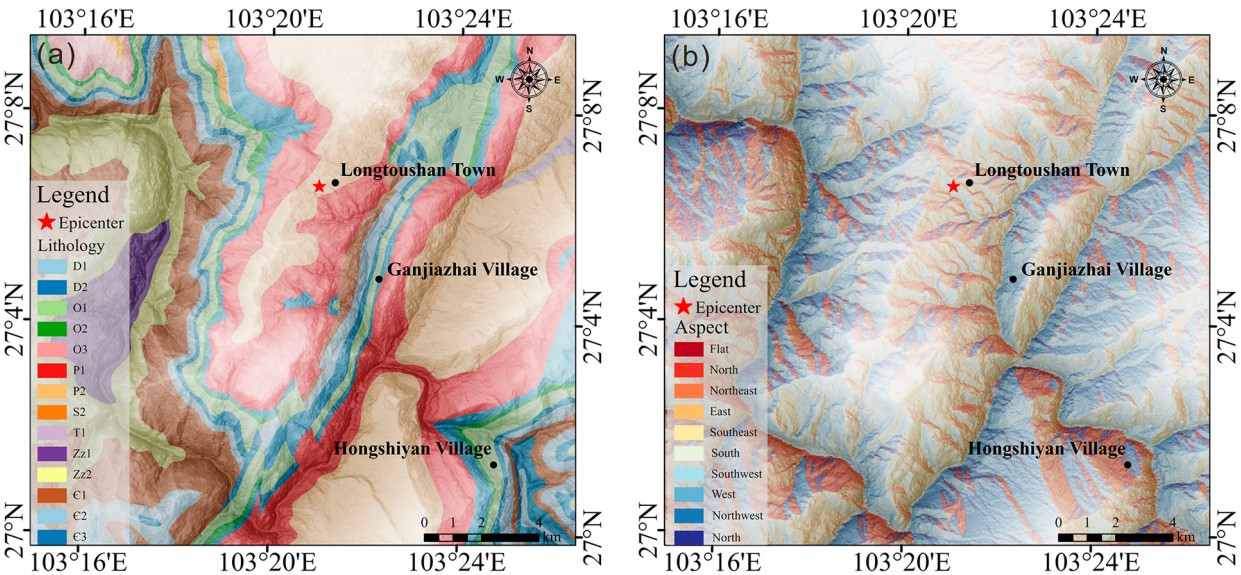

**Figure 8.** *Cont*.

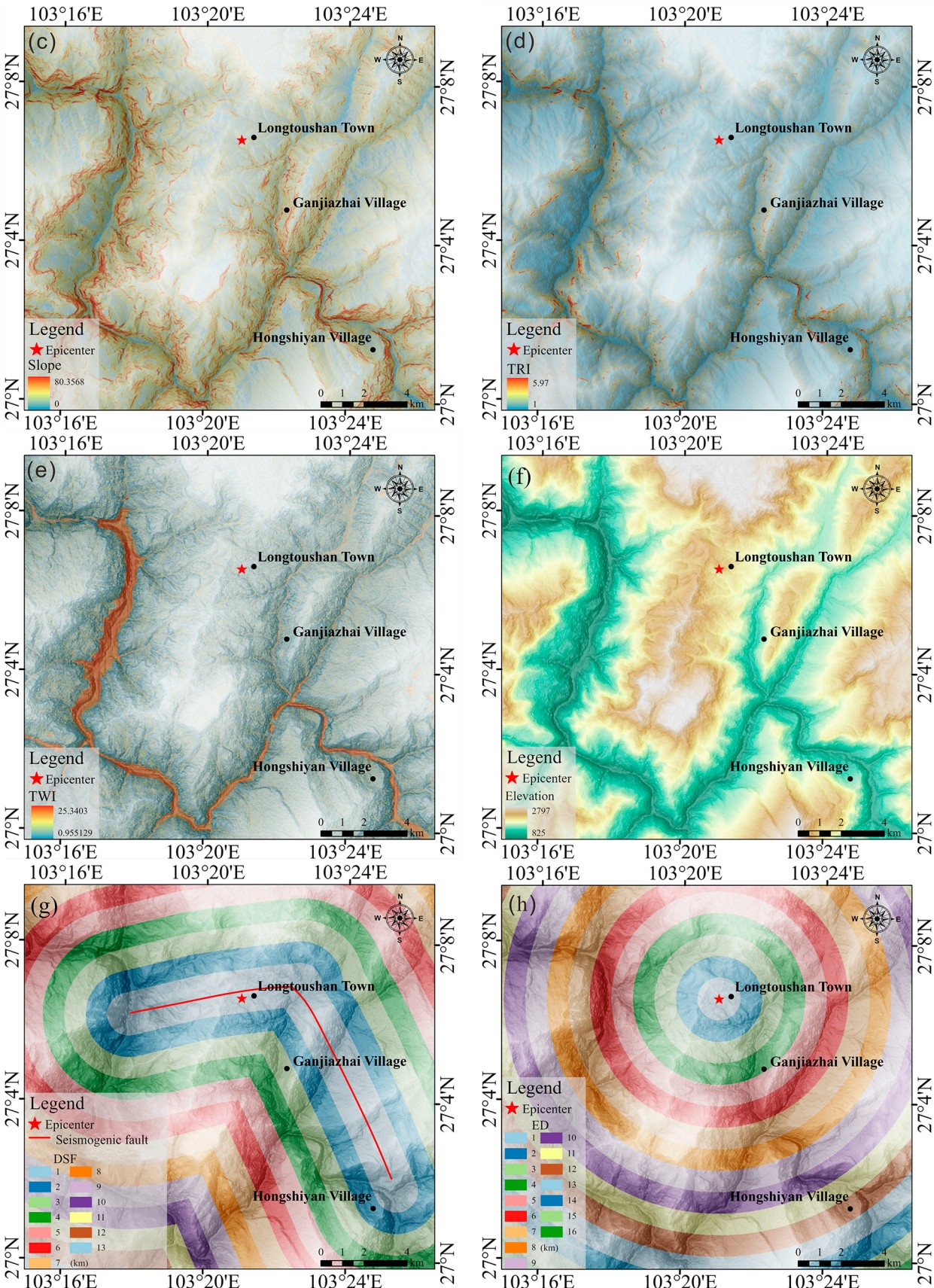

**Figure 8.** *Cont.*

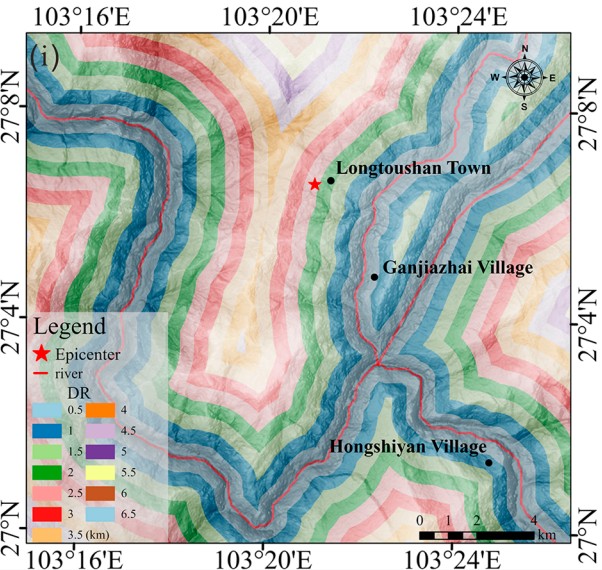

**Figure 8.** Raster maps of other factors: lithology (**a**), aspect (**b**), slope (**c**), TRI (**d**), TWI (**e**), elevation (**f**), DSF (**g**), ED (**h**), and DR (**i**).

$$V = \sqrt{\frac{x^2}{M min\{(r-1),(c-1)\}}} \tag{4}$$

where $x^2$ represents the Chi-square statistic, $M$ represents the total sample size, $r$ and $c$ represents the number of rows and columns, respectively.

The results of the correlation calculation between the controlling factors and landslide occurrence are shown in Figure 9. Therefore, we eliminated TWI from the selected factors due to its absolute correlation value being less than 0.1.

### 3.1.3. Correlation Analysis of Controlling Factors

To address the issue of parameter redundancy and mitigate model instability resulting from multicollinearity among factors, it was necessary to perform correlation analysis as a preprocessing step. Given the high correlation observed among the four ground motion parameters, this section will focus on analyzing the correlation specifically among these factors, using SED as an example. To visualize the correlations among the nine parameters, a correlation matrix and a schema ball were generated and are presented in Figure 10.

During factor correlation analysis, we observed the highest positive correlation coefficient (0.89) between slope and TRI. As a result, we excluded TRI from the final set of variables. Finally, we selected terrain factors (elevation, slope, aspect, DR), geological factor (lithology), and seismic factors (SED, ED, DSF) to analyze their impact on landslide distribution.

### 3.2. Model Strategy

The random forest algorithm, a robust machine learning technique consisting of multiple classification and regression trees, incorporates the bagging technique (bootstrap aggregation) to randomly select samples from the training dataset and construct classification and regression trees accordingly. It also identifies the best splits from random subsets of predisposing factors. The out-of-bag samples are used to evaluate the model's error. The outcome of the random forest combines the results from all classification and regression trees, leading to smoother and more consistent predicted values by mitigating the discontinuities often observed in individual trees. Despite the advantages of classification and regression tree algorithms, there are limitations as an individual tree that can potentially undermine their predictive accuracy. Firstly, the sensitivity of classification

and regression trees to the training dataset implies that variations in the training data may lead to significant differences in the constructed trees. Secondly, the constrained number of leaf nodes within the tree restricts the range of predicted values, resulting in predictions with discontinuities. To address these drawbacks, the random forest (RF) classifier was introduced [86].

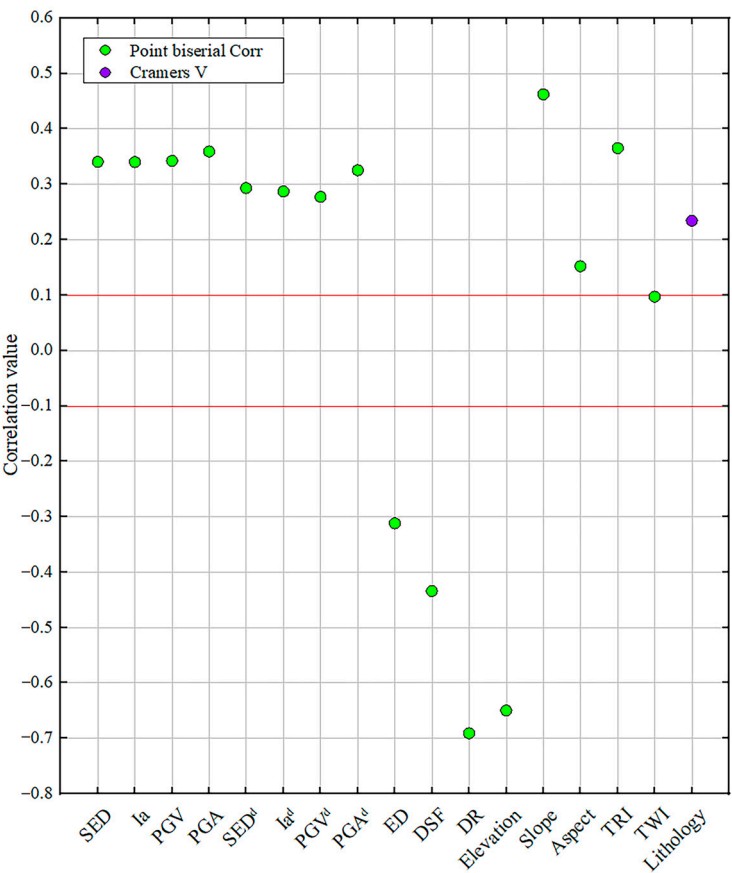

**Figure 9.** Correlation coefficients between controlling factors and landslide occurrence.

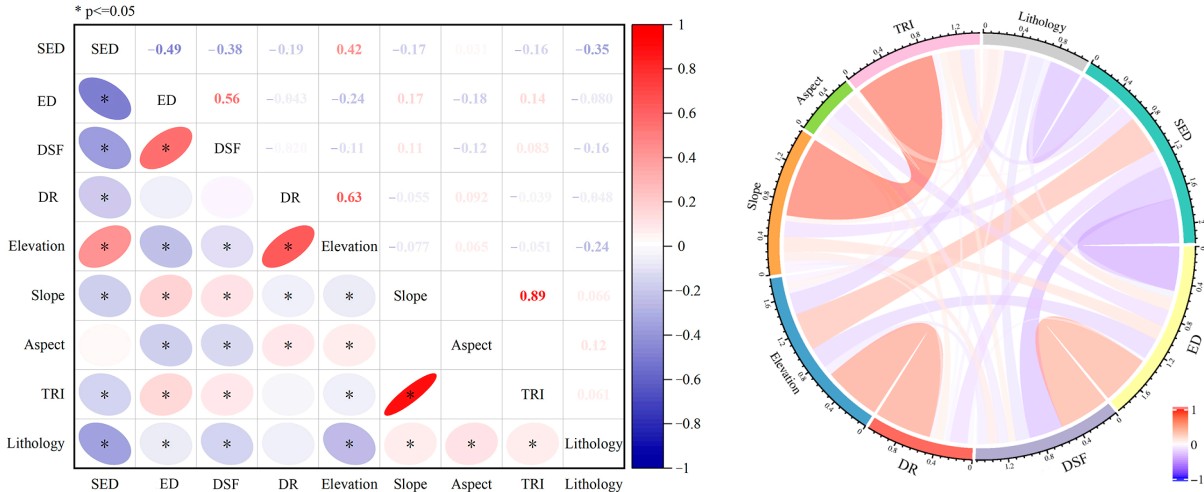

**Figure 10.** Correlations among the 9 factors.

Random forest algorithms have been successfully employed in several studies on landslide susceptibility evaluation [66]. These algorithms have been proved to possess the highest predictive capability and best performance compared to other shallow machine

learning algorithms [87]. Therefore, we used the random forest function in Python Studio to develop a model with the training dataset. This model enables us to calculate the landslide susceptibility values at each location in the study area using the characterized predisposing factors. To explore the effects of ground motion parameters on landslide distribution, we employed eight evaluation strategies by combining each ground motion parameter with seven other factors (Table 6).

**Table 6.** Combination strategy of eight evaluation models.

| Model | Model Formula |
|---|---|
| 1 | PGA + Elevation + Slope + Aspect + ED + DSF + DR + Lithology |
| 2 | PGV + Elevation + Slope + Aspect + ED + DSF + DR + Lithology |
| 3 | SED + Elevation + Slope + Aspect + ED + DSF + DR + Lithology |
| 4 | Ia + Elevation + Slope + Aspect + ED + DSF + DR + Lithology |
| 5 | $PGA^d$ + Elevation + Slope + Aspect + ED + DSF + DR + Lithology |
| 6 | $PGV^d$ + Elevation + Slope + Aspect + ED + DSF + DR + Lithology |
| 7 | $SED^d$ + Elevation + Slope + Aspect + ED + DSF + DR + Lithology |
| 8 | $Ia^d$ + Elevation + Slope + Aspect + ED + DSF + DR + Lithology |

We extracted eight factors for 1470 landslide samples and 1470 non-landslide samples, respectively. Subsequently, we employed all samples to train a random forest model with a training-to-testing sample ratio of 7:3. By performing a grid search, we determined the optimal model parameters that yielded the highest accuracy. Afterward, we randomly generated 3000 samples within the study area, considering the order of magnitude comparable to the samples used for model training. We extracted attributes from all these samples and utilized the trained model to predict outcomes for the 3000 samples. The predictions generated values ranging from 0 to 1.

## 4. Results

### 4.1. Landslide Susceptibility Mapping

Figure 11 displays the landslide susceptibility distribution associated with the eight evaluation strategies for the Ludian earthquake. The landslide susceptibility maps generated were categorized into five distinct classes using the Jenks natural breaks method: very low (0–0.15), low (0.15–0.35), medium (0.35–0.6), high (0.6–0.8), and very high (0.8–1). Upon visual inspection, the distribution patterns of landslide susceptibility generated from the eight strategies exhibit similarities, with high susceptibility areas concentrated along the Niulan, Longquan, and Shaba rivers. The distribution of landslide susceptibility resulting from the first four strategies, which do not consider the directional variation in ground motion parameters, shows similar patterns, with no significant differences in distribution and consistent area sizes among all categories. However, when the directional variation in ground motion parameters is considered, there are noticeable differences in the spatial distribution of landslide susceptibility. Specifically, the high susceptibility range becomes smaller, especially in the southwest region, while the low susceptibility range spreads out further. These changes in distribution are more consistent with the actual distribution pattern of the co-seismic landslides, indicating the importance of considering directional variation in accurately assessing landslide susceptibility.

### 4.2. Evaluation of the Models

#### 4.2.1. Model Performance

The Receiver Operating Characteristic (ROC) curve, which was introduced by Mandrekar [88], is a widely adopted statistical test to evaluate the accuracy of spatial prediction models. It measures the true-positive rates of selected spatial prediction models and is commonly used in landslide susceptibility mapping [89]. The Area Under the Curve (AUC) is commonly employed as a metric to evaluate the accuracy of the models. Higher AUC values indicate better accuracy, and those ranging from 0.5 to 1 are considered good ac-

curacy [90]. The ROC curves of the eight strategies indicate a high level of prediction accuracy for all models (Figure 12). However, the strategies that do not take into account the directional variation in ground motion parameters exhibit higher prediction accuracy, with AUC values as high as 0.93, compared to those that consider it. The AUC values of the last four models, which include SED[d], PGV[d], PGA[d], and Ia[d], are 0.89, 0.88, 0.86, and 0.85, respectively. In addition, regardless of whether the directional variation in ground motion parameter is considered or not, the models utilizing SED and PGV as ground motion parameters show better predictive ability and goodness-of-fit compared to those using Ia and PGA. In addition, the model performance between SED and PGV shows no significant difference.

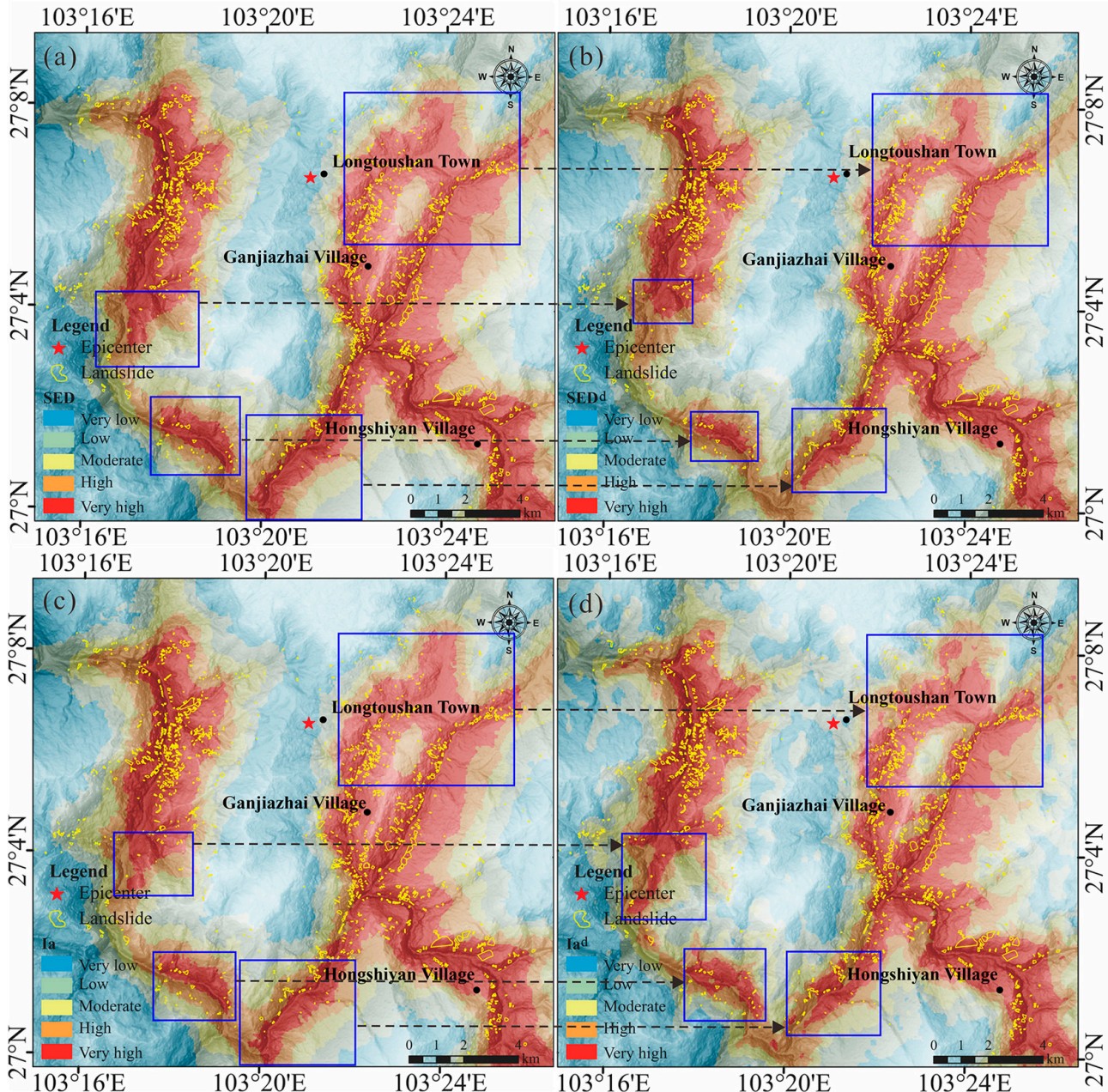

**Figure 11.** *Cont.*

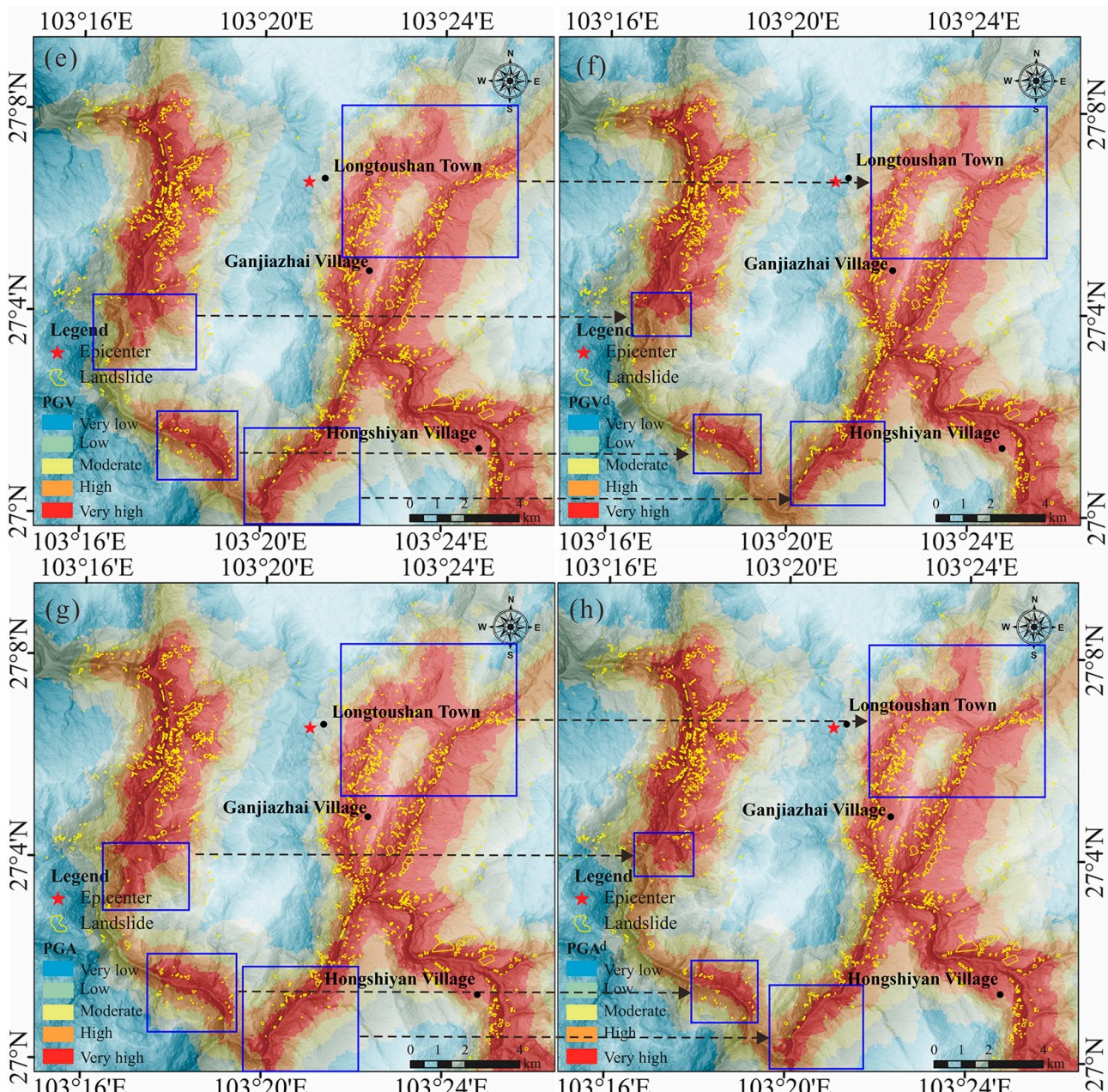

**Figure 11.** The landslide susceptibility maps obtained from the eight strategies incorporating the ground motion parameters are as follows: SED (**a**), SED$^d$ (**b**), Ia (**c**), Ia$^d$ (**d**), PGV (**e**), PGV$^d$ (**f**), PGA (**g**), and PGA$^d$ (**h**). The blue boxes highlight areas where significant changes occur in the landslide susceptibility maps when considering the directional variation in ground motion parameters.

### 4.2.2. Comparison of Predicted Areas

The actual distribution of co-seismic landslides highlights the importance of minimizing the total proportion of areas classified as "high" and "very high" in landslide susceptibility mapping. As shown in Figure 13 and Table 7, the evaluation model considering the directional variation in ground motion parameters exhibits a lower total proportion of areas susceptible to landslides as "high" and "very high", with the SED$^d$ model exhibiting the lowest value at 33.43%. In contrast, the models without considering the directional effect have a higher overall percentage of areas classified as "high" and "very high" severity, with the PGA model having the highest proportion at 37.33%. Among the eight strategies evaluated, the prediction results using SED$^d$ as the ground motion parameter demonstrate the lowest proportion of area

susceptible to co-seismic landslides of "high" and "very high" severity, with proportions of 19.2% and 14.23%, respectively. Such results show a decrease of 3.2% and 0.51%, respectively, compared to the prediction results obtained using SED as the ground motion parameter. This finding demonstrates that the landslide susceptibility distribution generated by this model is more consistent with the actual distribution of landslides.

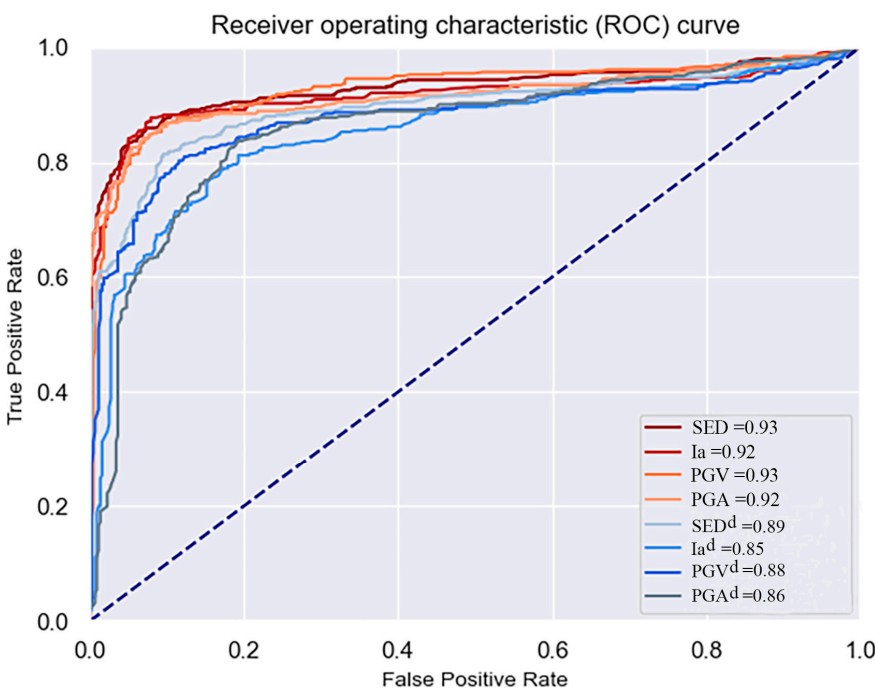

**Figure 12.** Performance of 8 strategies of landslide susceptibility model.

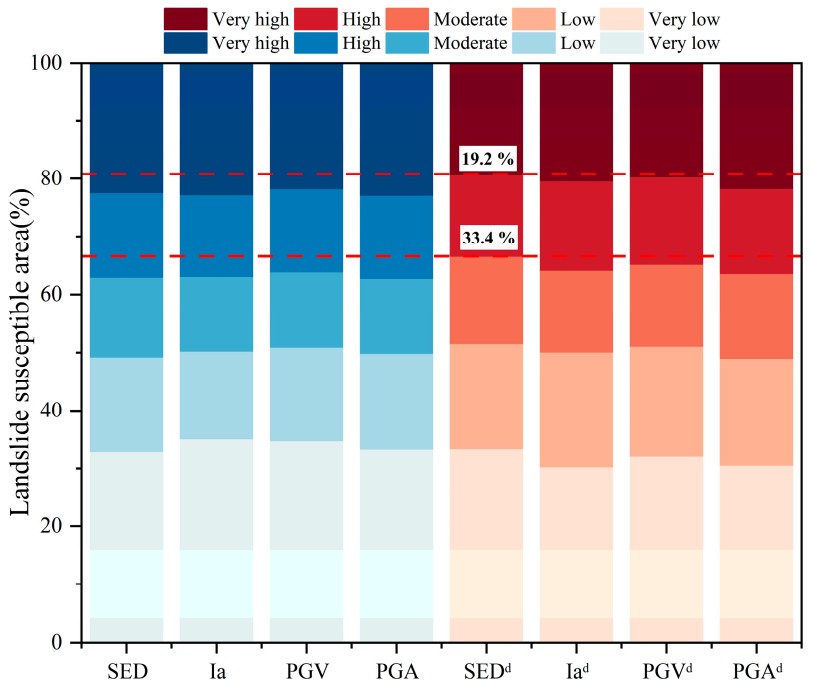

**Figure 13.** Landslide susceptible area by eight models: SED, Ia, PGV, PGA, SED$^d$, Ia$^d$, PGV$^d$, PGA$^d$.

### 4.3. The Importance of Controlling Factors

This study analyzed the reasons for the unusual spatial distribution pattern of Ludian co-seismic landslides from the perspective of controlling factors. The earlier discussion

mainly focused on the influence of ground motion parameters on the accuracy of spatial prediction models. In this section, we examine the combined influence of all factors on the distribution of landslides. Figure 14 displays the importance ranking of each factor in the prediction model. It is prominent that regardless of whether directional variation is considered or not, the ground motion parameter is not the most significant factor. The importance of DR and elevation is much greater than that of other factors, indicating that these two factors play a crucial controlling role in the spatial distribution of landslides. Furthermore, when the directional variation is considered, the importance of the ground motion parameter decreases, while the importance of DSF increases. This comprehensive analysis emphasizes the combined influence of seismologic, topographic, geomorphic, hydrological, and lithological factors in controlling the unusual spatial distribution pattern of the Ludian earthquake-triggered landslides.

**Table 7.** Area proportion of each susceptibility class.

| Scheme | Very Low | Low | Moderate | High | Very High | High + Very High |
|---|---|---|---|---|---|---|
| SED | 32.9 | 16.38 | 13.59 | 14.74 | 22.4 | 37.14 |
| Ia | 35.17 | 15.1 | 12.79 | 14.18 | 22.75 | 36.93 |
| PGV | 34.86 | 16.08 | 12.99 | 14.32 | 21.75 | 36.07 |
| PGA | 33.27 | 16.62 | 12.78 | 14.51 | 22.82 | 37.33 |
| SED[d] | 33.33 | 18.39 | 14.85 | 14.23 | 19.2 | 33.43 |
| Ia[d] | 30.28 | 19.84 | 14.11 | 15.38 | 20.4 | 35.78 |
| PGV[d] | 32.1 | 19.01 | 14.13 | 15.04 | 19.72 | 34.76 |
| PGA[d] | 30.57 | 18.46 | 14.62 | 14.66 | 21.69 | 36.35 |

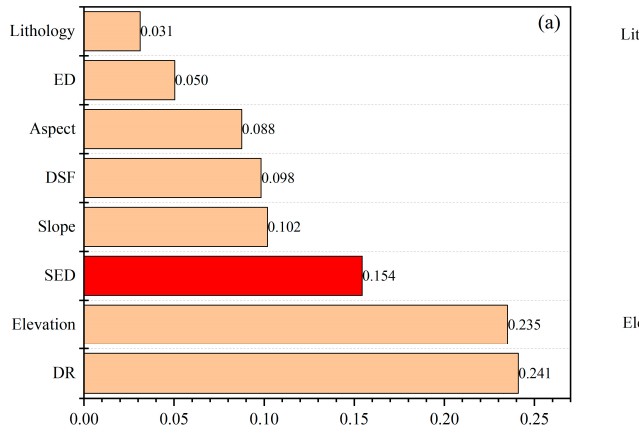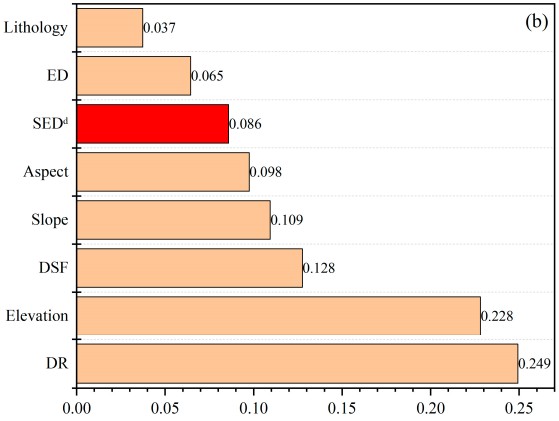

**Figure 14.** Ranking importance of controlling factors without (**a**) and with (**b**) direction consideration. The red bars indicate the importance of the ground motion parameter.

## 5. Discussion

Through landslide susceptibility mapping, this study provided new insights into the factors contributing to the unusual distribution pattern of the Ludian earthquake-triggered landslides. We focused on examining the influence of ground motion parameters on the likelihood of co-seismic landslide occurrence, as well as exploring the association between seismic energy variations in different directions and the corresponding degrees of slope instability. Based on the results, there are two noteworthy points for discussion: (1) the accuracy of the model and (2) the decreased importance of the ground motion parameter when the directional effect is considered.

### 5.1. The Completeness of the Landslide Inventory

Creating a comprehensive and detailed co-seismic landslide inventory is crucial for studying the spatial distribution of co-seismic landslides, assessing their susceptibility,

and understanding their impact on the geomorphic evolution of earthquake-affected areas. However, in this study, the resulting landslide inventory exhibits certain disparities in terms of the number of landslides, due to the relatively limited coverage and the specific criteria used for interpreting landslides with areas exceeding 500 m$^2$. It is important to note that despite these differences, the LAP obtained in our study (2.34%) is relatively close to the result of other more comprehensive inventories (2.71%). This similarity suggests that the landslide database we established possesses a certain level of representativeness and can be considered reliable for assessing co-seismic landslide susceptibility. Nonetheless, investigating the impact of inventory completeness, particularly in terms of the number of landslides, in subsequent research is recommended.

### 5.2. Strategies for Selecting Landslide Samples and Prediction Models

Extensive research on the selection strategy of landslide samples and models has been conducted over the past several years [90,91]. With the development of artificial intelligence, more scholars are using deep learning models with complex architectures. Dou et al. [92] analyzed the accuracy of four sampling strategies, (i.e., the centroid of landslide body, the landslide scarp centroid, samples of the landslide body, and samples of the scrap region) using logistic regression (LR), artificial neural network (NNET), and deep learning neural network (DNN) models, in order to perform the co-seismic landslide susceptibility mapping of the 2018 Hokkaido earthquake. Their results revealed that the most accurate strategy was landslide scarp, followed by landslide body, centroid of scarp, and centroid of body. However, the evaluation of the DNN model revealed that the accuracies of the four types of samples were largely similar, suggesting that the sampling strategy had little impact on the accuracy of DNN models with deep architecture. In contrast, Yi et al. [93] discovered that the accuracy of Convolutional Neural Network (CNN) models with the same deep architecture was influenced by the choice of sampling strategy at different scales, suggesting that the selection of samples and models remains a topic of ongoing debate within the field.

The present study employed the centroid of scarp as the sampling strategy, which has been broadly used in previous studies. The evaluation model selected for the study was random forest, which is also a well-established and extensively used model in landslide susceptibility assessments.

However, it is worth exploring different sampling strategies and deep learning models with complex architectures to gain a better understanding of the underlying causes of the unusual spatial distribution patterns of co-seismic landslides in the 2014 Ludian earthquake. Additionally, further analyses are required to investigate the possible reasons for the lower AUC of the model when the directional variation in ground motion parameter was taken into account.

### 5.3. Influence of Landslide Size

As mentioned earlier in this paper, the size of landslides triggered by the Ludian earthquake exhibit a distinct spatial distribution pattern. Hakan and Luigi [94] emphasized the importance of considering landslide size in co-seismic landslide assessments, a factor that has often been neglected in previous studies. Despite significant advancements in landslide susceptibility evaluation methods, ranging from traditional bivariate statistical models to deep learning models, there is still a lack of spatially explicit data-driven models that are capable of predicting the size of landslides in a specific study area. To address this problem, Lombardo et al. [95] first introduced a novel approach by employing a traditional statistical method called the Generalized Additive Model to predict the landslide area. This methodology fills a significant gap in the current literature as there is a lack of available maps that can statistically estimate the anticipated extent of slope failures. Following this study, Aguilera et al. [96] proposed the Hierarchical Neural Network (HNN), which is the first hierarchical data-driven model capable of simultaneously estimating both the potential locations of landslides and the corresponding size classes.

Incorporating landslide size into landslide susceptibility assessments requires the acquirement of detailed information on the area, depth, and volume of the landslides, which is challenging and may necessitate extensive fieldwork or remote sensing techniques. However, databases of global co-seismic landslides that encompass detailed landslide size information are limited. Although the emphasis of this study was placed on analyzing the spatial locations of the landslides rather than their sizes, obtaining size information on the landslides, particularly their volume, would be highly valuable for further investigating the influence of directional variation in ground motion parameters on the distribution of co-seismic landslides triggered by the Ludian earthquake.

## 6. Conclusions

Based on our investigation of the factors influencing the unusual landslide distribution caused by the 2014 Mw 6.2 Ludian earthquake, several major conclusions are obtained, which are outlined below:

(1) The Ludian earthquake-triggered landslides are not linearly concentrated along the seismogenic fault, but rather dispersed along major river systems with an NE–SW trend. The two most important factors that significantly affected the spatial distribution of these landslides were found to be the distance to rivers and elevation.

(2) The $R_i$ values for slopes facing SE, S, and SW are 1.3, 1.23, and 1.41, respectively, while slopes facing N, NW, and NE have much lower $R_i$ values (0.77, 0.68, and 0.52). Therefore, the percentage of landslide source area on the slopes facing south is much larger than that on the slopes facing north, which is consistent with seismic energy variations, i.e., the value of ground motion parameters in the south is larger than that in the north.

(3) The model's performance and its ability to accurately represent the spatial distribution of co-seismic landslides were essentially the same, regardless of whether the analysis incorporated PGA, PGV, Ia, or SED. However, in comparison to PGA[d], PGV[d] and Ia[d], SED[d] emerged as the most effective ground motion parameter for interpreting the distribution of co-seismic landslides.

(4) The occurrence of co-seismic landslides during the 2014 Ludian earthquake exhibits a significant relationship between the directional variation in ground motion parameters and different slope aspects. Although the AUC of the model slightly decreases when the directional variation in ground motion parameters is taken into account, there is a notable reduction in the proportion of areas of "high" and "very high" landslide susceptibility. This adjustment results in a better accordance between the model's prediction and the actual distribution of landslides. Therefore, we suggest that the directional variation in ground motion parameters plays an essential role and should be taken into account in the co-seismic landslide susceptibility mapping for the Ludian earthquake.

**Author Contributions:** All authors significantly contributed to the research. Conceptualization, J.L.; data curation, J.L. and Y.D.; formal analysis, J.L. and Y.D.; funding acquisition, J.L.; methodology, Y.D., J.L. and Z.L.; resources, X.P.; software, Y.D. and J.L.; supervision, X.P. and J.L.; visualization, Y.D.; Writing—Original draft, Y.D. and J.L.; Writing—Review and editing, J.L. and X.P. All authors have read and agreed to the published version of the manuscript.

**Funding:** This research was funded by [National Natural Science Foundation of China] grant number [42107212] and [Natural Science Foundation of Sichuan Province] grant number [2022NSFSC1145].

**Data Availability Statement:** Data are available upon request from the corresponding author.

**Acknowledgments:** We gratefully acknowledge the provision of high-quality seismic metadata from 28 strong motion stations by the Institute of Engineering Mechanics, China Earthquake Administration.

**Conflicts of Interest:** The authors declare no conflict of interest.

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
