# Peer review of "Co-Seismic Landslides Triggered by the 2014 Mw 6.2 Ludian Earthquake, Yunnan, China: Spatial Distribution, Directional Effect, and Controlling Factors"

_remotesensing, doi:10.3390/rs15184444_

Round 1

Reviewer 1 Report

The objective of this paper is to study the co-seismic landslides triggered by the 2014 Mw 6.2 Ludian earthquake (Yunnan, China), emphasizing on the spatial distribution, directional effect and controlling factors. In particular, the spatial distribution pattern of co-seismic landslides in the earthquake, were elucidate, while the ground motion parameter (among PGA, PGV, Ia, and SED) that most significantly contributes to co-seismic landslides in the earthquake was determined. Finally, the relationship between seismic energy variations in different directions and the corresponding varying degrees of slope instability was examined.

This is an interesting and well-structured paper, which includes all necessary sections (Introduction, Study area and Materials, Methodology, Results, Discussion, Conclusions) are included. Moreover, the “Study area and Materials”, “Methodology”, “Results” and “Discussion” sections are divided into several sub-sections, providing detailed descriptions. Furthermore, all Figures, Tables and Diagrams are consistent with the analysis described in the manuscript. Regarding the mathematical part, predominantly analyzed in the “Methodology” section, it is valid and accurately explained. However, some changes should be implemented, which will result in the paper improvement. Particularly:

Lines 9-31: Although the abstract has been properly structured, unnecessary details are contained (these details could be placed in the manuscript). The abstract should be clear and concise, while the most significant processes/findings/conclusions should be highlighted. Please, modify the abstract by reducing its length.

Line 42: After this sentence, I suggest adding a brief paragraph, in which global examples, related to landslides triggered by earthquakes, will be included. Indicative papers, in which the corresponding information can be obtained and can optionally be cited, are the following: 1. Qin, Y., Wei, J. B., Zheng, H. C., & Cui, Y. L. (2012). Cause and Stability Analysis of Daomakan Lanslide in Xiangjiaba Reservoir. Applied Mechanics and Materials, 204–208, 581–585. https://doi.org/10.4028/www.scientific.net/AMM.204-208.581, 2. Sboras, S., Lazos, I., Bitharis, S., Pikridas, C., Galanakis, D., Fotiou, A., Chatzipetros, A., & Pavlides, S. (2021). Source modelling and stress transfer scenarios of the October 30, 2020 Samos earthquake: seismotectonic implications. Turkish Journal of Earth Sciences, 30, 699–717. https://doi.org/10.3906/yer-2107-25, 3. Shao, X., & Xu, C. (2022). Earthquake-induced landslides susceptibility assessment: A review of the state-of-the-art. Natural Hazards Research, 2(3), 172–182. https://doi.org/10.1016/j.nhres.2022.03.002. Please, apply.

Line 140: After this paragraph, I suggest adding a new and brief paragraph, in which the seismicity of the study area will be further explained. Maybe, a map with the focal mechanisms, obtained from a catalog, for a specific period (e.g. last 20 years) could be added. Please, apply.

Line 186, 214, 216 and 250: Please, provide more detailed descriptions in the Figures 3, 4, 5 and 6 captions, respectively.

Author Response

We would like to thank you, most sincerely, for all the effort and expertise that you have contributed to reviewing. Your valuable comments are great helpful to improve the manuscript. We have carefully considered the comments and have revised the manuscript accordingly. We hope that the revised manuscript is acceptable. Detailed responses to your comments are given below.

M1. Lines 9-31: Although the abstract has been properly structured, unnecessary details are contained (these details could be placed in the manuscript). The abstract should be clear and concise, while the most significant processes/findings/conclusions should be highlighted. Please, modify the abstract by reducing its length.

Based on your suggestions, we have streamlined the abstract section to make it more focused on the most important processes/findings/conclusions of this study. (i.e., The 2014 Mw 6.2 Ludian earthquake exhibited a structurally complex source rupture process and an unusual spatial distribution pattern of co-seismic landslides. In this study, we constructed a spatial database consisting 1470 co-seismic landslides, each exceeding 500 m2. These landslides covered a total area of 8.43 km2 and were identified through a comprehensive interpretation of high-resolution satellite images taken before and after the earthquake. It is noteworthy that the co-seismic landslides do not exhibit a linear concentration along the seismogenic fault; instead, they predominantly extend along major river systems with an NW-SE trend. Moreover, the southwest-facing slopes have the highest landslide area ratio of 1.41. To evaluate the susceptibility of the Ludian earthquake-triggered landslides, we performed a random forest model that considered topographic factors (elevation, slope, aspect, distance to rivers), geological factor (lithology), and seismic factors (ground motion parameters, epicentral distance, distance to the seismogenic fault). Our analysis revealed that the distance to rivers and elevation were the primary factors influencing the spatial distribution of the Ludian earthquake-triggered landslides. When we considered the directional variation of ground motion parameters, the AUC of the model slightly decreased. However, incorporating this variation led to a significant reduction in the proportion of areas classified as "high" and "very high" landslide susceptibility. Moreover, SEDd emerged as the most effective ground motion parameter for interpreting the distribution of the co-seismic landslides when compared to PGAd, PGVd and Iad.).

M2. Line 42: After this sentence, I suggest adding a brief paragraph, in which global examples, related to landslides triggered by earthquakes, will be included. Indicative papers, in which the corresponding information can be obtained and can optionally be cited, are the following:1. Qin, Y., Wei, J. B., Zheng, H. C., & Cui, Y. L. (2012). Cause and Stability Analysis of Daomakan Lanslide in Xiangjiaba Reservoir. Applied Mechanics and Materials, 204–208, 581–585. https://doi.org/10.4028/www.scientific.net/AMM.204-208.581, 2. Sboras, S., Lazos, I., Bitharis, S., Pikridas, C., Galanakis, D., Fotiou, A., Chatzipetros, A., & Pavlides, S. (2021). Source modelling and stress transfer scenarios of the October 30, 2020 Samos earthquake: seismotectonic implications. Turkish Journal of Earth Sciences, 30, 699–717. https://doi.org/10.3906/yer-2107-25. 3. Shao, X., & Xu, C. (2022). Earthquake-induced landslides susceptibility assessment: A review of the state-of-the-art. Natural Hazards Research, 2(3), 172–182. https://doi.org/10.1016/j.nhres.2022.03.002. Please, apply.

Following your suggestion, we have added a description of the global example of earthquake-triggered landslides in Lines 34–38. (i.e., Over the past decades, numerous studies have been conducted on co-seismic landslides in mountainous areas, such as the 2004 Mw 6.6 mid-Niigata earthquake [13], the 2005 Mw 7.6 Northern Pakistan earthquake [14], the 2008 Ms 8.0 Wenchuan earthquake [15], the 2018 Mw 6.6 Hokkaido Eastern Earthquake [16], the 2020 Mw 6.9 Samos earthquake [17].) And the paper titled “Earthquake-induced landslides susceptibility assessment: A review of the state-of-the-art”, which provides an overview of the state-of-the-art techniques for earthquake-induced landslides susceptibility assessment, was quoted in Line 107.

M3. Line 140: After this paragraph, I suggest adding a new and brief paragraph, in which the seismicity of the study area will be further explained. Maybe, a map with the focal mechanisms, obtained from a catalog, for a specific period (e. g. last 20 years) could be added. Please, apply.

Based on your suggestion, we have incorporated the location, time, and magnitude of historical earthquakes in Figure 1. However, in order to emphasize the seismic core area, we had to limit the view scope, which resulted in not being able to display information for all 115 historical earthquakes in Figure 1. Instead, we have provided a description of historical earthquakes within the study area in Lines 142–147 (i.e., A total of 115 earthquakes with magnitudes equal to or greater than 4.7, occurring between 624 CE and 2014 CE within a 200-kilometer radius around the epicenter of the 2014 Ludian main shock, were extracted from the China Earthquake Administration. It is evident that while the Ludian earthquake is classified as a moderate event, it is also considered a low-frequency seismic occurrence in this region, with an estimated recurrence interval of about 100 years [52]).

M4. Line 186, 214, 216 and 250: Please, provide more detailed descriptions in the Figures 3, 4, 5 and 6 captions, respectively.

In response to your suggestion, we have added caption descriptions for Figures 3, 4, 5, and 6 (i.e., Figure 3. Density distribution of co-seismic landslides in the Ludian earthquake. The ellipses are the standard deviation ellipse plotted with the number of landslides as weight. Figure 4. Area distribution of co-seismic landslides in the Ludian earthquake. The ellipses are the standard deviation ellipse plotted with the landslide source area as weight. Figure 5. The LAP-LND distribution of the Ludian co-seismic landslides. The provenance areas of landslides are divided into square grids of 0.1 km2 each. Figure 6. Relationship between the abundances of co-seismic landslide source areas and slope aspect: 0° (North), 90° (East), 180° (South), 270° (West).).

Reviewer 2 Report

This paper uses a random forest model for statistical correlation analysis to determine the relationship between the spatial distribution of landslides and various factors. Terrain factors (elevation, slope, aspect), hydrological factors (distance to rivers), geological factors (lithology), and seismic factors (seismic motion parameters, epicenter distance, distance to seismic faults) are selected to evaluate the sensitivity of landslides caused by the Ludian earthquake. It was found that co seismic landslides do not distribute linearly along seismic faults, but mainly extend along the main river system with a northwest southeast trend. The author has obtained many interesting distribution patterns and features of landslides through statistical methods. However, the article lacks an attempt to analyze the mechanisms behind these laws and characteristics from the perspective of landslide formation mechanisms. It is hoped that the author can supplement the investigation and analysis, and increase relevant analysis and discussion.

Other comments:

How negative sample data is generated and what is its basis. The author's introduction to the generation of negative sample data is too brief. The generation of negative samples is a key step in landslide disaster assessment and is related to the reliability of the entire study. Therefore, it is recommended that the author supplement the relevant discussion on negative sample generation, including the commonly used methods in landslide negative sample generation in current research. The author needs to explain the reasons for the method used in this article and discuss the reliability of the selected method.

The font size in Figure 2 is not coordinated, and the longitude and latitude fonts are too large. This problem also exists in other images. What does non landslide sampling area in the figure mean. Why is there a clear boundary between landslide areas and non landslide areas, and why cannot non landslide points be selected in landslide areas?

When analyzing the spatial distribution of landslides, the author discovered the distribution pattern of landslides, but did not provide relevant analysis to explain the reasons for the distribution pattern. The author found that co seismic landslides are not linearly distributed along seismic faults or clustered around the epicenter area. On the contrary, landslides are mainly distributed along rivers. Has the author investigated the reasons for the distribution of these landslides along rivers and provided reasonable explanations?

What is the basis for eliminating factors with a correlation less than 0.1 with landslide occurrence.

Table 4 provides abbreviations for many factors, but there is no explanation or basis for selecting these factors in the text.

The Discussion section did not fully discuss the limitations of this study, as well as the lack of analysis of the distribution mechanism of landslides. The full text only analyzed the distribution and characteristics of landslides from the aspects of distribution and statistical laws, lacking an internal explanation based on the formation mechanism of landslides, which cannot bring readers deeper thinking and understanding.

Author Response

We would like to thank you, most sincerely, for all the effort and expertise that you have contributed to reviewing. Your valuable comments are great helpful to improve the manuscript. We have carefully considered the comments and have revised the manuscript accordingly. We hope that the revised manuscript is acceptable. Detailed responses to your comments are given below.

General comments:

This paper uses a random forest model for statistical correlation analysis to determine the relationship between the spatial distribution of landslides and various factors. Terrain factors (elevation, slope, aspect), hydrological factors (distance to rivers), geological factors (lithology), and seismic factors (seismic motion parameters, epicenter distance, distance to seismic faults) are selected to evaluate the sensitivity of landslides caused by the Ludian earthquake. It was found that co seismic landslides do not distribute linearly along seismic faults, but mainly extend along the main river system with a northwest southeast trend. The author has obtained many interesting distribution patterns and features of landslides through statistical methods. However, the article lacks an attempt to analyze the mechanisms behind these laws and characteristics from the perspective of landslide formation mechanisms. It is hoped that the author can supplement the investigation and analysis, and increase relevant analysis and discussion.

M1. How negative sample data is generated and what is its basis. The author's introduction to the generation of negative sample data is too brief. The generation of negative samples is a key step in landslide disaster assessment and is related to the reliability of the entire study. Therefore, it is recommended that the author supplement the relevant discussion on negative sample generation, including the commonly used methods in landslide negative sample generation in current research. The author needs to explain the reasons for the method used in this article and discuss the reliability of the selected method.

In response to your suggestion, we have provided a detailed description of the methodology, process, and reliability of the landslide negative sample generation in Lines 208–230 (i.e., To ensure a balanced model, an equal number of negative samples were generated in our study. There are various methods available for generating negative samples. Selecting negative samples from areas with lower landslide occurrence probabilities is a valuable approach that can greatly enhance the reliability of landslide susceptibility prediction [16, 18]. Two commonly used methods for negative sample selection are random sampling [74, 75] and buffer-controlled sampling [76, 77]. However, these methods have a limitation in that they cannot guarantee the selection of non-landslide samples from areas with extremely low and low susceptibility levels. To overcome this limitation, the Information Value model [78, 79] and the Mean Clustering model [80] have been preliminarily applied to study of landslide negative sample selection. The Fuzzy c-means algorithm (FCM) possesses the capability to classify the study area into different levels of landslide susceptibility based on the geographical attributes of landslide controlling factors. This categorization allows for the generate non-landslide samples from areas with low susceptibility levels. Notably, FCM is not influenced by subjective factors and operates independent of any specific model. In a comparative analysis conducted by Liang [80] on selecting non-landslide samples for assessing shallow landslide susceptibility using machine learning, FCM outperformed the K-means algorithm in terms of sampling reliability. Consequently, this study has chosen to utilize the FCM for the extraction of non-landslide samples. We employed FCM to categorize landslide susceptibility in the study area into five classes, i.e., “Very low”, “Low”, “Moderate”, “High”, and “Very high” (Figure 2). The “Very low” and “Low” areas were selected as non-landslide sampling areas, from which a total of 1470 negative samples were randomly generated. This approach ensures a representative set of non-landslide samples for our analysis.).

M2. The font size in Figure 2 is not coordinated, and the longitude and latitude fonts are too large. This problem also exists in other images. What does non landslide sampling area in the figure mean. Why is there a clear boundary between landslide areas and non landslide areas, and why cannot non landslide points be selected in landslide areas?

Based on your suggestion, we have made modifications to the font size of latitude and longitude in Figures 2, 7, and 8.

Selecting negative samples from areas with lower landslide occurrence probabilities is a valuable approach that can greatly enhance the reliability of landslide susceptibility prediction (Liu et al., 2021; Xiong et al., 2022). Two commonly used methods for negative sample selection are random sampling (Pourghasemi et al., 2018; Bordoni et al., 2020) and buffer-controlled sampling (Nefeslioglu et al., 2008; Zhu et al., 2019). However, these methods have a limitation in that they cannot guarantee the selection of non-landslide samples from areas with extremely low and low susceptibility levels. To overcome this limitation, the Information Value model (Zhou et al., 2023; Huang et al., 2023) and the Mean Clustering model (Liang, 2022) have been preliminarily applied to study of landslide negative sample selection. The Fuzzy c-means algorithm (FCM) possesses the capability to classify the study area into different levels of landslide susceptibility based on the geographical attributes of landslide controlling factors. This categorization allows for the generate non-landslide samples from areas with low susceptibility levels. Notably, FCM is not influenced by subjective factors and operates independent of any specific model. In a comparative analysis conducted by Liang (2022) on selecting non-landslide samples for assessing shallow landslide susceptibility using machine learning, FCM outperformed the K-means algorithm in terms of sampling reliability. Consequently, this study has chosen to utilize the FCM for the extraction of non-landslide samples. Please refer to the relevant information in Lines 208–230.

Liu Y, Zhang W, Zhang Z. Risk factor detection and landslide susceptibility mapping using Geo-Detector and Random Forest Models: The 2018 Hokkaido eastern Iburi earthquake. Remote Sensing 2021, 13(6): 1157.

Xiong, Y.B.; Zhou, Y.; Wang, F.T.; Wang, S.X. A Novel Intelligent Method Based on the Gaussian Heatmap Sampling Technique and Convolutional Neural Network for Landslide Susceptibility Mapping. Remote Sens. 2022, 14(12), 2866.

Pourghasemi, H.R.; Rahmati, O. Prediction of the landslide susceptibility: Which algorithm, which precision? Catena 2018, 162: 177-192.

Bordoni, M.; Galanti, Y.; Bartelletti, C. The influence of the inventory on the determination of the rainfall-induced shallow landslides susceptibility using generalized additive models. Catena 2020, 193: 104630.

Nefeslioglu, H.A.; Gokceoglu, C.; Sonmez, H. An assessment on the use of logistic regression and artificial neural networks with different sampling strategies for the preparation of landslide susceptibility maps. Eng. Geol. 2008, 97(3-4): 171-191.

Zhu, A.X.; Miao, Y.; Liu, J. A similarity-based approach to sampling absence data for landslide susceptibility mapping using data-driven methods. Catena 2019, 183, 104188.

Zhou, C.; Gan, L.L.; Wang, Y. Regional landslide susceptibility modeling with integrated non-landslide sample selection in-dex and heterogeneous integrated machine learning. J. Geo-Info. Sci. 2023, 25(08), 1570-1585. (In Chinese)

Huang, F.M.; Zeng, S.Y.; Huang, J.S. Uncertainty in predictive modeling of landslide susceptibility: the effect of different “non-landslide sample” selection methods. Eng. Sci. Technol. 2023, 1, 1-14. (In Chinese)

Liang, Z. Integrated application and research of machine learning in shallow landslide sensitivity evaluation[D]. 2022. (In Chinese)

M3. When analyzing the spatial distribution of landslides, the author discovered the distribution pattern of landslides, but did not provide relevant analysis to explain the reasons for the distribution pattern. The author found that co seismic landslides are not linearly distributed along seismic faults or clustered around the epicenter area. On the contrary, landslides are mainly distributed along rivers. Has the author investigated the reasons for the distribution of these landslides along rivers and provided reasonable explanations?

Relevant reason analysis has been added in Lines 245–258. (i.e., When investigating the underlying formation mechanism behind this specific distribution pattern, it's essential to consider the following factors. Firstly. almost 60 % of the slopes in the study area ranges from 10° to 30°, while slopes steeper than 40° are mainly distributed along the Niulan, Shaba and Longquan rivers. The topographic amplification of seismic responses significantly increases the susceptibility of these steeper slopes to instability. Secondly, Due to the process of valley incision, the slopes along the riverbanks exhibit steep terrain and undergo substantial weathering and unloading. Consequently, shallow landslides, which represent the primary type of landslides induced by the Ludian earthquake [58], are highly prone to occur. Lastly, the proximity of the rivers to the epicenter and seismogenic fault implies that the intense shaking also affects the slopes along the riverbanks. However, in areas closer to the epicenter and seismogenic fault, the landscape is predominantly characterized by gentle slopes. This suggests that more intense shaking is require to initiate sliding and there is relative stability under dynamic seismic conditions. This finding coincides with the results revealed by Chen et al. [81].).

M4. What is the basis for eliminating factors with a correlation less than 0.1 with landslide occurrence.

We apologize for any confusion caused by the clerical errors in the manuscript. Based on your suggestion, the phrase "We excluded factors with correlations less than 0.1" has been revised to "Therefore, we eliminated TWI from the selected factors due to its absolute correlation value being less than 0.1" in the revision.

M5. Table 4 provides abbreviations for many factors, but there is no explanation or basis for selecting these factors in the text.

The reasonable selection of controlling factors significantly impacts the accuracy and authenticity of landslide susceptibility assessment results. However, there is currently no unified standard for choosing controlling factors. Ayalew et al. (2005) suggested that the criteria for selecting controlling factors should be based on their similarity to landslide locations, measurability, non-redundancy, and knowledge of the geo-environmental conditions of the study area. Therefore, taking guidance from the previous research on the co-seismic landslide susceptibility assessment (Chang et al., 2021; Shao and Xu, 2022; Lee et al., 2008; Parise and Jibson, 2000), we carefully identified and selected 10 factors that are strongly associated with the occurrence of co-seismic landslides, considering three key perspectives: seismic, topographic, and geological aspects. Subsequently, we carried out correlation analyses to examine the relationship between landslides and controlling factors in section 3.1.2. Additionally, we conducted individual correlation analyses among the controlling factors themselves in section 3.1.3. Based on the results of these correlation analyses, we excluded TWI and TRI from the final set of variables. Finally, we selected terrain factors (elevation, slope, aspect, DR), geological factor (lithology), and seismic factors (SED, ED, DSF) to analyze their impact on landslide distribution. And the high accuracy of spatial prediction models based on these factors is also evident the reasonably choose of the controlling factors. The utilization of these factors in our analysis has led to the development of spatial prediction models with a notably high level of accuracy in section 4.2.1. This outcome underscores the appropriateness of our selection when it comes to the controlling factors for our study on landslide distribution.

Ayalew, L.; Yamagishi, H. The application of GIS-based logistic regression for landslide susceptibility mapping in the Kaku-da-Yahiko Mountains, Central Japan. Geomorphology 2005, 65, 15–31.

Chang, M.; Zhou, Y.; Zhou, C.; Hales, T.C. Coseismic landslides induced by the 2018 Mw 6.6 Iburi, Japan, Earthquake: spatial distribution, key factors weight, and susceptibility regionalization. Landslides 2021, 18, 755–772.

Shao, X.; Xu, C. Earthquake-induced landslides susceptibility assessment: A review of the state-of-the-art. Nat. Hazards Res. 2022, 2(3), 172–182.

Lee, C.T.; Huang, C.C.; Lee, J.F. Statistical approach to earthquake-induced landslide susceptibility. Eng. Geol. 2008, 100(1-2): 43-58.

Parise, M; Jibson, R.W. A seismic landslide susceptibility rating of geologic units based on analysis of characteristics of landslides triggered by the 17 January, 1994 Northridge, California earthquake. Eng. Geol. 2000, 58(3-4): 251-270.

M6. The Discussion section did not fully discuss the limitations of this study, as well as the lack of analysis of the distribution mechanism of landslides. The full text only analyzed the distribution and characteristics of landslides from the aspects of distribution and statistical laws, lacking an internal explanation based on the formation mechanism of landslides, which cannot bring readers deeper thinking and understanding.

The limitations of this study have been further discussed in the revision, focusing on the following aspects: the completeness of the landslide inventory, strategies for selecting landslide samples and prediction models, and influence of landslide size. These discussions provide specific directions to enhance the universality and authority of the conclusions drawn in this study.

The analysis of the distribution mechanism of landslides has been added in Lines 245–258. (i.e., When investigating the underlying formation mechanism behind this specific distribution pattern, it's essential to consider the following factors. Firstly. almost 60 % of the slopes in the study area ranges from 10° to 30°, while slopes steeper than 40° are mainly distributed along the Niulan, Shaba and Longquan rivers. The topographic amplification of seismic responses significantly increases the susceptibility of these steeper slopes to instability. Secondly, Due to the process of valley incision, the slopes along the riverbanks exhibit steep terrain and undergo substantial weathering and unloading. Consequently, shallow landslides, which represent the primary type of landslides induced by the Ludian earthquake [58], are highly prone to occur. Lastly, the proximity of the rivers to the epicenter and seismogenic fault implies that the intense shaking also affects the slopes along the riverbanks. However, in areas closer to the epicenter and seismogenic fault, the landscape is predominantly characterized by gentle slopes. This suggests that more intense shaking is require to initiate sliding and there is relative stability under dynamic seismic conditions. This finding coincides with the results revealed by Chen et al. [81].).

Reviewer 3 Report

Brief summary:

The manuscript studies the correlation of topographic, seismic, hydrographic, and geological factors with the occurrence of landslides triggered by the 2014 Ludian earthquake. It is found that the distance to rivers and the surface elevation have the highest correlations with the occurrence of landslides. A random forest model is applied to investigate the influence of ground motion parameters on the prediction of co-seismic landslides. It is found that specific energy density (SED) and peak ground velocity (PGV) provide better performance than peak ground acceleration (PGA) or arias density (Ia). Controversial results are obtained when the directional variation of ground motion parameters is taken into account.

General comments:

1) The manuscript improves current knowledge by studying the relationship between SED and the occurrence of co-seismic landslides. However, SED and the other ground motion parameters PGA, PGA, and Ia have high correlation between each other as can be seen from Fig. 7, correlate quite similarly with the occurrence of landslides as shown in Fig. 9, and have quite similar ROC curves in Fig. 12. Consequently, it seems that the results do not improve essentially when using SED instead of the other ground motion parameters, which have been studied a lot by others in [19-29] as cited in the introduction.

2) The manuscript is well-organized and fluent to read. The figures are visually of high quality, although Figs. 11 and 12 are not as sharp as the other ones. The approach is appropriate while mistakes appear in the details.

Detailed comments:

Lines 13-14 and Table 1: UAV images have been mentioned only in the abstract but not elsewhere in the manuscript. 

Lines 20-25: The random forest model was not used in the correlation analysis but to study the influence of the ground motion parameters. The correlation analysis was based on the point biserial correlation coefficient and the Cramer's V.

Line 28 and Fig. 9: According to Fig. 9, PGA has the highest correlation with landslide occurrence and not SED among the ground motion parameters.

Lines 69-70: The reference to [48-40] seems inappropriate. Do you mean [38-40]?

Line 71: The abbreviation DDA has not been explained.

Line 127: An average slope of 1.22% sounds small. Does it mean the slope of the river or the mountains?

Line 158: Please provide more details about the identification of landslides from the satellite images. Were they identified manually or automatically before field verification?

Line 167-168: Please clarify how the landslide susceptibility was defined in the FCM algorithm.

Figure 4 and lines 160 and 194: The largest landslide was 0.6 km^2 according to line 160, which is larger than 0.2 km^2 stated on line 194 and drawn in Fig. 4.

Line 198 and Fig. 5: It is not clear which value has been used for the total area to end up with the LAP values given in the table in Fig. 5.

Lines 218-228: LAD has not been defined. Do you mean LAP? It is not clear to which the SDE of LAD (or LAP) is compared to. If the SDE of LAP is compared to the SDE of LND, then the flatness increases and not decreases. No centripetal force is associated with either of the SDEs.

Figure 6: The figure does not really show a correlation but R_i as a function of the slope aspect. R_{ij} = R_{ii} = R_i(aspect). R_{ij} for i not equal j have not been shown, so the notation R_{ij} is inappropriate. 

Lines 251-257 and Table 3: For slopes facing W, the ratio A_i/A = 0.138772 which is closer to 15% than 11%. R_i rounds to 1.23 and not 1.22 for slopes facing S. R_i for slopes facing NW is also low in addition to slopes facing N and NE.

Lines 278 and 299-301: It would better to explain the abbreviations TRI and TWI already on line 278, where they appear for the first time.

Equation 1: For clarity, it would be better to denote the variable of integration by a different symbol than t, which is the upper bound of the integral (SED = SED(t)).

Equation 2: It is not necessary to give the unit (m/s) since no units appear in other equations either.

Lines 294-297: It is not clear how the mosaicing approach was carried out. Are the direction-dependent raster maps combinations of all directions? What kind of mosaicing method was applied?

Lines 297-298 and Figs. 1 and 7: There are a few seismic stations visible in Fig. 1, one of which is located in the study area. Please explain how the dense raster maps of PGA, PGV, Ia, and SED were generated from observations at sparse locations. Were they interpolated?

Figure 8a: The lithology labels which appear in the legend have not been explained.

Line 310: A continuity factor has not been defined. Do you mean the controlling factor?

Lines 323-324 and Fig. 9. There are several negative correlation values below 0.1, which have not been omitted. It seems that factors for which the absolute value of the correlation is below 0.1 are omitted.

Lines 330-331: The conclusion that the ground motion factors correlated because they were calculated from the same data source is unjustified. According to Table 4, there were also other factors which were calculated from the same data source (either China Earthquake Administration or ALOS DEM) and which do not correlate. The correlations depend on the variables and not on the data source.

Lines 347-355: Please clarify that the limitations concern individual trees and not the random forest.

Lines 360-362 and 372: Please clarify how the susceptibility is defined in the random forest model. It ranges from 0 to 1, so is it equal to LAP, which is further classified into the five classes?

Lines 363-364 and Table 5: There are seven and not eight factors in addition to the ground motion parameter, which is the eighth factor.

Lines 380-382: The reduction of high susceptibility areas for directional variables in Fig. 11 is apparently related to a similar effect visible in Fig. 7.

Figure 11 caption: The maps have not been produced by the ground motion factors only but by the different strategies involving eight factors each.

Line 410: Total ratio seems inappropriate. Do you mean total proportion?

Line 468: The abbreviation CNN has not been explained.

Line 502: Area information was available as described in Sections 2.2, 2.3.2, and Table 3. Volume information may be missing.

Lines 514-515: The same comment as for lines 251-257. R_i rounds to 1.23 and not 1.22 for slopes facing S. R_i for slopes facing NW is also low in addition to slopes facing N and NE.

Lines 520-521 and 404-405: The performance was essentially the same for the model which included PGV as for the model which included SED, so SED was not the only one which provided the highest performance. Moreover, no results were shown for a model without any ground motion parameter, so it cannot be concluded that incorporating a ground motion parameter improved the performance.

Author Response

We would like to thank you, most sincerely, for all the effort and expertise that you have contributed to reviewing. Your valuable comments are great helpful to improve the manuscript. We have carefully considered the comments and have revised the manuscript accordingly. We hope that the revised manuscript is acceptable. Detailed responses to your comments are given below.

General comments:

1) The manuscript improves current knowledge by studying the relationship between SED and the occurrence of co-seismic landslides. However, SED and the other ground motion parameters PGA, PGA, and Ia have high correlation between each other as can be seen from Fig. 7, correlate quite similarly with the occurrence of landslides as shown in Fig. 9, and have quite similar ROC curves in Fig. 12. Consequently, it seems that the results do not improve essentially when using SED instead of the other ground motion parameters, which have been studied a lot by others in [19-29] as cited in the introduction.

The motivation for conducting this study originated from a curiosity about a specific phenomenon observed from the co-seismic landslides and the directional differences of seismological signals in the Ludian earthquake. Specifically, the largest landslide triggered by the Ludian earthquake was the Hongshiyan landslide, with a volume of approximately 12.24 Mm3. Interestingly, the opposite slope, which has a similar geologic structure but steeper topography, does not show obvious deformation associated with the earthquake. Moreover, the values of the PGA, PGV, Ia, and SED in the direction facing the Hongshiyan slope are significantly larger than those in the opposite slope. Therefore, we used a random forest model to conduct co-seismic landslide hazard assessment. The primary objectives of the study are as follows: (1) to elucidate the spatial distribution pattern of co-seismic landslides in the earthquake, an earthquake caused by a complex faulting process; (2) to determine the ground motion parameter that most significantly contributes to co-seismic landslides in the earthquake among PGA, PGV, Ia, and SED; (3) to examine the relationship between seismic energy variations in different directions and the corresponding varying degrees of slope instability.

Our analysis revealed that the distance to rivers and elevation were the primary factors influencing the spatial distribution of the Ludian earthquake-triggered landslides. The model's performance and its ability to accurately represent the spatial distribution of co-seismic landslides were essentially the same, regardless of whether the analysis incorporated PGA, PGV, Ia, or SED. Although the AUC of the model slightly decreases when the directional variation of ground motion parameters is taken into account, there is a notable reduction in the proportion of areas of ”high” and ”very high” landslide susceptibility. Therefore, it is suggested that the directional variation of ground motion parameters plays an essential role and should be taken into account in the co-seismic landslide susceptibility mapping for the Ludian earthquake. Moreover, in comparison to PGAd, PGVd and Iad, SEDd emerged as the most effective ground motion parameter for interpreting the distribution of the co-seismic landslides.

The research results largely align with our initial hypotheses. However, as you have pointed out, PGA, PGA, Ia, SED show high correlation with each other, as depicted in Figure 7, and demonstrate similar correlations with the occurrence of landslides, as shown in Figure 9. Consequently, their AUC values in Figure 12 are quite similar. Nevertheless, it is important to note that there is a lack of studies specifically investigating the relationship between co-seismic landslides and SED, despite the strong evidence indicating a positive correlation between SED and the intensity of earthquake damage. Moreover, further analysis is urgently needed to enhance the universality and authority of the findings. This could involve addressing the limitations of this study, as discussed in the section of Discussion, and exploring other earthquakes that exhibit similar phenomena, such as the 2018 Hokkaido earthquake and the 2016 Kumamoto earthquake, among others.

2) The manuscript is well-organized and fluent to read. The figures are visually of high quality, although Figs. 11 and 12 are not as sharp as the other ones. The approach is appropriate while mistakes appear in the details.

Based on your suggestions, we have improved the quality of Figures 11 and 12 to make them visually clearer.

M1. Lines 13-14 and Table 1: UAV images have been mentioned only in the abstract but not elsewhere in the manuscript.

We have included details regarding UAV images in Lines 169 and Table 1, in response to your recommendations. (i.e., These images comprised Sentinel-2A images (10 m resolution), GF-1 images (2 m resolution), GF-2 images (1 m resolution), Google Earth data (0.5 m resolution) and UAV (0.2 m resolution) (Table 1).)

M2. Lines 20-25: The random forest model was not used in the correlation analysis but to study the influence of the ground motion parameters. The correlation analysis was based on the point biserial correlation coefficient and the Cramer's V.

Taking into account your valuable suggestion, we have made revisions to the manuscript, ensuring an enhanced content that aligns effectively with your feedback. Please refer to the relevant revisions mentioned in Lines 16–19 (i.e., To evaluate the susceptibility of the Ludian earthquake-triggered landslides, we performed a random forest model that considered topographic factors (elevation, slope, aspect, distance to rivers), geological factor (lithology), and seismic factors (ground motion parameters, epicentral distance, distance to the seismogenic fault).).

M3. Line 28 and Fig. 9: According to Fig. 9, PGA has the highest correlation with landslide occurrence and not SED among the ground motion parameters.

Yes, as shown in Figure 9, the PGA exhibits the highest correlation with landslide occurrence among the ground motion parameters, rather than the SED. However, Figure 12 indicated that the model's performance and its ability to accurately represent the spatial distribution of co-seismic landslides were essentially the same, regardless of whether the analysis incorporated PGA, PGV, Ia, or SED. Moreover, the occurrence of co-seismic landslides during the 2014 Ludian earthquake exhibits a significant relationship between the directional variation of ground motion parameters and different slope aspects. Although the AUC of the model slightly decreases when considering the directional variation of ground motion parameters, there is a notable reduction in the proportion of areas of ”high” and ”very high” landslide susceptibility. SEDd emerged as the most effective ground motion parameter for interpreting the distribution of the co-seismic landslides when compared to PGAd, PGVd and Iad. Therefore, relevant sentence has been revised in Lines 21–26 (i.e., When we considered the directional variation of ground motion parameters, the AUC of the model slightly decreased. However, incorporating this variation led to a significant reduction in the proportion of areas classified as "high" and "very high" landslide susceptibility. Moreover, SEDd emerged as the most effective ground motion parameter for interpreting the distribution of the co-seismic landslides when compared to PGAd, PGVd and Iad.).

M4. Lines 69-70: The reference to [48-40] seems inappropriate. Do you mean [38-40]?

We have adjusted the order of the references.

M5. Line 71: The abbreviation DDA has not been explained.

We have incorporated the complete name of DDA, which stands for Discontinuous Deformation Analysis, in Lines 70–71.

M6. Line 127: An average slope of 1.22% sounds small. Does it mean the slope of the river or the mountains?

"1.22% average slope" refers to the average slope of the river, not the average slope of the mountains. To accurately reflect this information, we have modified the content in Lines 127–128 as follows: "The study area features a "V-shaped" high mountain valley landscape with an average stream slope of 1.22% and a natural drop of approximately 220 m.”.

M7. Line 158: Please provide more details about the identification of landslides from the satellite images. Were they identified manually or automatically before field verification?

Before conducting field verification, we adopted manual interpretation based on satellite images. This approach was chosen because automated methods can occasionally lead to an overestimation of total landslide volumes when they inadvertently combine nearby individual events into a single entity. The detailed information on the manual process of identifying landslides from satellite images have been included in Lines 163–183 (i.e., Landslide inventories play a crucial role in analyzing the spatial distribution of landslides, evaluating their causative failure mechanisms, and generating susceptibility maps. The interpretation of the 2014 Ludian earthquake-triggered landslides was conducted by using an extensive collection of high-resolution satellite images captured before and after the earthquake. These images comprised Sentinel-2A images (10 m resolution), GF-1 images (2 m resolution), GF-2 images (1 m resolution), Google Earth data (0.5 m resolution) and UAV (0.2 m resolution) (Table 1). All images were subjected to geometric correction, enhancement, and coordinate system conversion. In order to address the potential overestimation of total landslide volumes that can occur with automated methods when nearby individual events are inadvertently combined into a single entity, we opted for manual digitization over automated extraction techniques to identify landslides in this study. The boundaries of landslides were manually digitalized into polygons using satellite images within a GIS environment. To guarantee the accuracy of the landslide inventory derived from remote sensing interpretation, we conducted an extensive field investigation within the densely populated area affected by co-seismic landslides, lasting for approximately 30 days. Based on the analysis of satellite images and the findings from the field investigations, we identified a total of 1470 landslides larger than 500 m2 within a rectangular area of approximately 360 km2. The total area covered by these landslides is estimated to be around 8.43 km2. The landslide number density (LND) obtained in this study is 1470/360 km2 = 4.083/km2 and the landslide area percentage (LAP) is (8.43 km2/360 km2) x 100% = 2.34%.).

M8. Line 167-168: Please clarify how the landslide susceptibility was defined in the FCM algorithm.

We have responded to your suggestion by incorporating a comprehensive description of the methodology, process, and reliability of the landslide negative sample selection in Lines 208–230. (i.e., To ensure a balanced model, an equal number of negative samples were generated in our study. There are various methods available for generating negative samples. Selecting negative samples from areas with lower landslide occurrence probabilities is a valuable approach that can greatly enhance the reliability of landslide susceptibility prediction [16, 18]. Two commonly used methods for negative sample selection are random sampling [74, 75] and buffer-controlled sampling [76, 77]. However, these methods have a limitation in that they cannot guarantee the selection of non-landslide samples from areas with extremely low and low susceptibility levels. To overcome this limitation, the Information Value model [78, 79] and the Mean Clustering model [80] have been preliminarily applied to study of landslide negative sample selection. The Fuzzy c-means algorithm (FCM) possesses the capability to classify the study area into different levels of landslide susceptibility based on the geographical attributes of landslide controlling factors. This categorization allows for the generate non-landslide samples from areas with low susceptibility levels. Notably, FCM is not influenced by subjective factors and operates independent of any specific model. In a comparative analysis conducted by Liang [80] on selecting non-landslide samples for assessing shallow landslide susceptibility using machine learning, FCM outperformed the K-means algorithm in terms of sampling reliability. Consequently, this study has chosen to utilize the FCM for the extraction of non-landslide samples. We employed FCM to categorize landslide susceptibility in the study area into five classes, i.e., “Very low”, “Low”, “Moderate”, “High”, and “Very high” (Figure 2). The “Very low” and “Low” areas were selected as non-landslide sampling areas, from which a total of 1470 negative samples were randomly generated. This approach ensures a representative set of non-landslide samples for our analysis.).

M9. Figure 4 and lines 160 and 194: The largest landslide was 0.6 km2 according to line 160, which is larger than 0.2 km2 stated on line 194 and drawn in Fig. 4.

We apologize for any confusion caused by the lack of clarity in the manuscript. The statement "the largest covers an area of approximately 0.6 km2" in the manuscript refers to the total area affected by the landslide, including its source area, circulation area, and accumulation area. On the other hand, the mentioned area of 0.2 km2 in Line 268 of revision specifically refers to the source area of the landslide. In the revision, relevant sentence has been revised to “The provenance area of the largest landslide is approximately 0.2 km2.”

M10. Line 198 and Fig. 5: It is not clear which value has been used for the total area to end up with the LAP values given in the table in Fig. 5.

In Figure 5, a total area of 4.28km2 represents the combined area of the 1470 landslide source areas. In the revision, we have made the necessary adjustments to provide a clearer description of the total area of the landslide and the provenance area of the landslide in Lines 180–181 (i.e., The total area covered by these landslides is estimated to be around 8.43 km2) and Lines 323–324 (i.e., the total landslide source-occupying area in the study area (4.28 km2)), respectively. Additionally, we have redrawn Figure 5 to reflect these revised descriptions accurately. Furthermore, we have introduced pertinent clarification that all the mentioned landslide areas specifically refer to the landslide source areas in Lines 206–207 (i.e., In addition, it is crucial to emphasize that all the landslide areas mentioned below specifically referring to the landslide source areas.).

M11. Lines 218-228: LAD has not been defined. Do you mean LAP? It is not clear to which the SDE of LAD (or LAP) is compared to. If the SDE of LAP is compared to the SDE of LND, then the flatness increases and not decreases. No centripetal force is associated with either of the SDEs.

Apologies for the clerical error. The term "LAD" was a mistake and has been corrected to "LAP". The primary purpose of conducting standard deviation ellipse analysis in Section 2.3.3 is to examine the variations in the spatial distribution characteristics of Ludian landslides from the perspectives of both landslide number density and landslide source area density. Therefore, during the construction of the two standard deviation ellipses using ArcGIS, we specifically selected landslide count and landslide area as the weighting factors. Taking into account the observation that the significant landslides triggered by the Ludian earthquake are predominantly distributed in the southeast direction of the epicenter, we observed a decrease in the flatness of the standard deviation ellipse when examining the spatial distribution characteristics based on the density of landslide source areas and using landslide source area as the weighting factor.

M12. Figure 6: The figure does not really show a correlation but R_i as a function of the slope aspect. R_{ij} = R_{ii} = R_i(aspect). R_{ij} for i not equal j have not been shown, so the notation R_{ij} is inappropriate.

We have made the necessary revision in the caption of Figure 6 by replacing the term "correlation" with "relation," (i.e., Relation of co-seismic landslide area abundances with slope aspect: 0° (North), 90° (East), 180° (South), 270° (West).). Moreover, we have made the appropriate adjustment by replacing “Rij” with “Ri” in Figure 6.

 M13. Lines 251-257 and Table 3: For slopes facing W, the ratio A_i/A = 0.138772 which is closer to 15% than 11%. R_i rounds to 1.23 and not 1.22 for slopes facing S. R_i for slopes facing NW is also low in addition to slopes facing N and NE.

Apologies for the confusion. We have now corrected the errors in Lines 326–332 as per your request (i.e., The study area exhibits a generally uniform slope aspect distribution, with the slopes facing SE, NW and W accounting for approximately 15% of the total area, while the remaining slope aspects account for approximately 11%. The  values for slopes facing SE, S, and SW are 1.30, 1.23, and 1.41, respectively. In contrast, the  values for other slope aspects are less than or close to 1. Particularly, the  values for N-, NW- and NE-oriented slopes are as low as 0.77, 0.68, and 0.52, respectively, indicating a low susceptibility to landslides in these areas.).

M14. Lines 278 and 299-301: It would better to explain the abbreviations TRI and TWI already on line 278, where they appear for the first time.

We have provided an explanation for the abbreviations TRI (Topographic Roughness Index) and TWI (Topographic Wetness Index), which were mentioned in Lines 356–357.

M15. Equation 1: For clarity, it would be better to denote the variable of integration by a different symbol than t, which is the upper bound of the integral (SED = SED(t)).

Following your suggestion, we have replaced t with T in Equation 1. (i.e., ).

M16. Equation 2: It is not necessary to give the unit (m/s) since no units appear in other equations either.

Based on your suggestion, we have modified Equation 2. (i.e., ).

M17. Lines 294-297: It is not clear how the mosaicing approach was carried out. Are the direction-dependent raster maps combinations of all directions? What kind of mosaicing method was applied?

The mosaic was accomplished using the raster mosaic tool in ArcGIS. We combined raster maps from four directions through mosaicking. Please refer to the relevant information in Lines 374–378 (i.e., Subsequently, the ground motion factor maps considering the directional effects were combined using the raster mosaic tool in ArcGIS. This process generated raster maps of direction-dependent PGA, PGV, SED, and Ia, which were abbreviated as PGAd, PGVd, SEDd, and Iad, respectively (Table 5).).

M18. Lines 297-298 and Figs. 1 and 7: There are a few seismic stations visible in Fig. 1, one of which is located in the study area. Please explain how the dense raster maps of PGA, PGV, Ia, and SED were generated from observations at sparse locations. Were they interpolated?

In this study, data from a total of 28 earthquake stations were used. However, considering the wide distribution of these stations, it was determined that displaying all of them on the map would adversely impact the visual representation of the earthquake core area. Therefore, only a subset of seismic stations is shown in Figure 1 and Figure 7. The source of the ground motion parameters and the methodology for generating the ground motion factor raster maps of PGA, PGV, Ia, and SED are described in Lines 361–364 (i.e., The ground motion parameters, including PGA, PGV, Ia, and SED, were derived from the high-quality metadata from 28 strong motion seismic stations provided by the China Earthquake Administration. To generate raster maps of the ground motion parameters, Kriging interpolation was employed.).

M19. Figure 8a: The lithology labels which appear in the legend have not been explained.

In the revision, we have included Table 4, which provides an explanation of the lithology labeling in Figure 8a.

M20. Line 310: A continuity factor has not been defined. Do you mean the controlling factor?

The term "continuous factors" refers to controlling factors, including Elevation, Aspect, Slope, TRI, TWI, ED, DSF, DR, PGA, PGV, SED, Ia, PGAd, PGVd, SEDd, and Iad, which possess attributes of continuous data. In contrast, the term “nominal factors” refers to factors with attributes of nominal data, specifically referring to lithology. The distinction between these two types of factors is important because the analysis of correlation between continuous factors and landslide incidence requires different methods compared to the analysis involving nominal factors. Please refer to the relevant information in Lines 395–403.

M21. Lines 323-324 and Fig. 9. There are several negative correlation values below 0.1, which have not been omitted. It seems that factors for which the absolute value of the correlation is below 0.1 are omitted.

I apologize for any confusion caused by the clerical errors in the manuscript. Based on your suggestion, the phrase "We excluded factors with correlations less than 0.1" has been revised to " we eliminated TWI from the selected factors due to its absolute correlation value being less than 0.1." in the revision.

M22. Lines 330-331: The conclusion that the ground motion factors correlated because they were calculated from the same data source is unjustified. According to Table 4, there were also other factors which were calculated from the same data source (either China Earthquake Administration or ALOS DEM) and which do not correlate. The correlations depend on the variables and not on the data source.

Based on your suggestion, we have revised relevant sentence in Lines 417–419 (i.e., Given the high correlation observed among the four ground motion parameters, this section will focus on analyzing the correlation specifically among these factors using SED as an example.).

M23. Lines 347-355: Please clarify that the limitations concern individual trees and not the random forest.

As per your request, we have made revisions to the content in Lines 435–442 (i.e., Despite the advantages of classification and regression tree algorithms, there are limitations as an individual tree that can potentially undermine their predictive accuracy. Firstly, the sensitivity of classification and regression trees to the training dataset implies that variations in the training data may lead to significant differences in the constructed trees. Secondly, the constrained number of leaf nodes within the tree restricts the range of predicted values, resulting in predictions with discontinuities. To address these drawbacks, the random forest (RF) classifier was introduced [87].).

M24. Lines 360-362 and 372: Please clarify how the susceptibility is defined in the random forest model. It ranges from 0 to 1, so is it equal to LAP, which is further classified into the five classes?

We have added the process for calculating landslide susceptibility using a random forest model in Lines 452–459. (i.e., We extracted eight factors for 1470 landslide samples and 1470 non-landslide samples, respectively. Subsequently, we employed all samples to train a random forest model with a training-to-testing sample ratio of 7:3. By performing a grid search, we determined the optimal model parameters that yielded the highest accuracy. Afterward, we randomly generated 3000 samples within the study area, considering the order of magnitude comparable to the samples used for model training. We extracted attributes from all these samples and utilized trained model to predict outcomes for the 3000 samples. The predictions generated values ranging from 0 to 1.).

Furthermore, it is important to note that the landslide susceptibility calculated by the Random Forest model differs from the results obtained through the LAP, which is a statistical analysis specifically focused on the area of co-seismic landslides. In contrast, the Random Forest model evaluates the susceptibility of the entire study area to landslides. There is no direct relationship between the outputs of the Random Forest model and the LAP.

M25. Lines 363-364 and Table 5: There are seven and not eight factors in addition to the ground motion parameter, which is the eighth factor.

Have revised. Please refer to the relevant information in Lines 449–451. (i.e., To explore the effects of ground motion parameters on landslide distribution, we employed eight evaluation strategies by combining each ground motion parameter with other seven factors (Table 6))

M26. Lines 380-382: The reduction of high susceptibility areas for directional variables in Fig. 11 is apparently related to a similar effect visible in Fig. 7.

Indeed, a clear correlation can be observed between the landslide susceptibility distribution depicted in Figure 11 and raster maps of ground motion parameters shown in Figure 7. By taking the direction variation of ground motion parameters into account, the raster maps of ground motion parameters provide a more accurate representation of the real seismic impact on the slope. Consequently, in this scenario, the landslide susceptibility distribution aligns more closely with the actual distribution of landslides. Therefore, we suggest that the directional variation of ground motion parameters plays an essential role and should be taken into account in the co-seismic landslide susceptibility mapping for the Ludian earthquake.

M27. Figure 11 caption: The maps have not been produced by the ground motion factors only but by the different strategies involving eight factors each.

The caption of Figure 11 has been revised to “The landslide susceptibility maps obtained from the eight strategies incorporating the ground motion parameters are as follows: SED (a), SEDd (b), Ia (c), Iad (d), PGV (e), PGVd (f), PGA (g), and PGAd (h)”.

M28. Line 410: Total ratio seems inappropriate. Do you mean total proportion?

Yes, we have changed “total ratio” to “total proportion”.

M29. Line 468: The abbreviation CNN has not been explained.

We have added the full name of CNN (Convolutional Neural Network) in Line 575.

M30 Line 502: Area information was available as described in Sections 2.2, 2.3.2, and Table 3. Volume information may be missing.

We have changed "area" to "volume" in Line 613.

M31. Lines 514-515: The same comment as for lines 251-257. R_i rounds to 1.23 and not 1.22 for slopes facing S. R_i for slopes facing NW is also low in addition to slopes facing N and NE.

In response to your suggestion, we have made modifications to the content of Lines 624–628 (i.e., The  values for slopes facing SE, S, and SW are 1.3, 1.23, and 1.41, respectively, while slopes facing N, NW and NE have much lower  values (0.77, 0.68, and 0.52). Therefore, the percentage of landslide area on slopes facing south is much larger than that on slopes facing north, which is consistent with seismic energy variations, i.e., the value of ground motion parameters in the south is larger than that in the north.)

M32. Lines 520-521 and 404-405: The performance was essentially the same for the model which included PGV as for the model which included SED, so SED was not the only one which provided the highest performance. Moreover, no results were shown for a model without any ground motion parameter, so it cannot be concluded that incorporating a ground motion parameter improved the performance.

In response to your suggestion, we have made modifications to the content of Lines 629–633 (i.e., (3) The model's performance and its ability to accurately represent the spatial distribution of co-seismic landslides were essentially the same, regardless of whether the analysis incorporated PGA, PGV, Ia, or SED. However, in comparison to PGAd, PGVd and Iad, SEDd emerged as the most effective ground motion parameter for interpreting the distribution of the co-seismic landslides.).

Reviewer 4 Report

The manuscript conducts research on landslides triggered by the Ludian earthquake, introduces the spatial distribution, directionality, and controlling factors of interpretation, and particularly emphasizes the impact of directionality on EQ-triggered landslide mapping. Indeed, in terms of the distribution characteristics of landslides within a small area of the Ludian earthquake, there is indeed an obvious correlation with the near-SN trend of the Niulan River and its tributaries. Here are some key questions for the author's reference:

1. To the number of landslides analyzed, the number of landslides used in this paper is 1,470, but some other scholars have interpreted more than 10,000 landslides. It is recommended that the co-authors analyze the completeness of the number of landslides and compare it with the results of previous studies during your revision job;

2. To the distribution of coseismic landslides, it is recommended that the author add a complete statement of the interpretation results instead of expressing the relative size of the area because the area and nature of the landslide greatly impact the analysis. The co-authors also pointed out the distinction between landslides, according to the reviewer's work experience in the research area, the central point of the provenance area in the study area is more accurate as the input of statistical analysis than the central point of the interpreted area of the entire landslide. Because this earthquake occurred in the deep-cut canyon area, the source area of the collapse is the key point of the landslide. As for the circulation area and accumulation area, they can change ever-changing according to the difference in terrain. This is a key point that needs to be emphasized;

3. In Figure 8, the co-authors quoted the previous research results of the earthquake surface rupture zone, but the buffer zone did not follow the scope of the rupture zone. It should be adjusted.  Although the SRL article (Xu et al., 2015) has been published, it is recommended that the co-authors mainly analyze it based on the seismic motion parameters, the re-located of the epiecenter, and the topographical parameters. According to the reviewer’s research results (unpublished), this time The earthquake may be a deep rupture, the surface is shaken and deformed, and the non-fracture cracks to the surface. If it must be analyzed, the relationship between the dominant direction of aftershocks and the spatial distribution of earthquakes can be analyzed;

4. Due to the relatively small-size magnitude of the Ludian earthquake and the small range of high-intensity areas, in terms of scale effects, the landform of the deep canyon at the epicenter determines the dominant direction of landslides, and the analysis results of earthquake cases are not universal. Meaning, once the landform structure changes, the so-called directionality will change;

5. Other small problems:

a) The nature of the fracture in Line 149 Figure 1 should be marked in the legend, the first letter of the name of the fracture is capitalized, and consistent, the source of the intensity map, and the source of the fracture should be marked;

b) Line 249 Figure 6 emphasizes the statistical relationship between the landslide area and the slope where it is located. It has been pointed out earlier that the provenance area of the landslide is the most important statistical source, and the accumulation area of the landslide cannot also be included in the statistical scope;

c) It is suggested that the Discussion section should add the impact of landslide data integrity on the analysis results, especially after the distinction of landslide provenance areas.

Author Response

We would like to thank you, most sincerely, for all the effort and expertise that you have contributed to reviewing. Your valuable comments are great helpful to improve the manuscript. We have carefully considered the comments and have revised the manuscript accordingly. We hope that the revised manuscript is acceptable. Detailed responses to your comments are given below.

M1. To the number of landslides analyzed, the number of landslides used in this paper is 1,470, but some other scholars have interpreted more than 10,000 landslides. It is recommended that the co-authors analyze the completeness of the number of landslides and compare it with the results of previous studies during your revision job.

Following your advice, we extensively referenced the research on Ludian earthquake landslide interpretation by Wu et al. (2020). In the revision, we compared the landslide interpretation results obtained in our study with those of their study. Based on the analysis of landslide area density, we have demonstrated that the landslide database created in this study is not only representative but also suitable for developing co-seismic landslides susceptibility of the 2014 Ludian earthquake. Please refer to the relevant revisions mentioned in Lines 184–195 (i.e., After the 2014 Ludian earthquake, some researches have interpreted the co-seismic landslides. The most comprehensive interpretation of landslides to date was performed by Wu et al. [71] in 2020, which yielded a landslide number of 12,817. They also compared with other results in terms of quality and resolution of remote sensing images, coverage, interpretation method, landslide number, and landslide area, indicating that their results are more complete, detailed and objective. The coverage area interpreted in their study was larger than that of our study. Additionally, in our study, all the interpreted landslide targets had an area greater than 500 m2. Despite these differences, it is noteworthy that the LAP obtained in our study (2.34 %) is relatively close to the result of their study (2.71 %). This similarity suggests that the landslide database we established possesses a certain level of representativeness and can be considered reliable for assessing co-seismic landslide susceptibility.).

In addition, in the discussion section, we have added the limitation of this study regarding the completeness of the landslide inventory.

Wu, W.Y.; Xu, C.; Wang, X.Q.; Tian, Y.Y.; Deng, F. Landslides Triggered by the 3 August 2014 Ludian (China) M w 6.2 Earthquake: An Updated Inventory and Analysis of Their Spatial Distribution. J. Earth Sci. 2020, 31, 853-866.

M2. To the distribution of coseismic landslides, it is recommended that the author add a complete statement of the interpretation results instead of expressing the relative size of the area because the area and nature of the landslide greatly impact the analysis. The co-authors also pointed out the distinction between landslides, according to the reviewer's work experience in the research area, the central point of the provenance area in the study area is more accurate as the input of statistical analysis than the central point of the interpreted area of the entire landslide. Because this earthquake occurred in the deep-cut canyon area, the source area of the collapse is the key point of the landslide. As for the circulation area and accumulation area, they can change ever-changing according to the difference in terrain. This is a key point that needs to be emphasized.

Based on your suggestions, we have included comprehensive information regarding the data sources, methodologies, and procedural steps employed for the interpretation of landslides. Furthermore, we have provided a detailed description of the basis and process involved in identifying the source areas of landslides. Please refer to the relevant revisions mentioned in Lines 196–207 (i.e., Since the primary focus of landslide susceptibility analysis revolves around the identification of potential landslide source areas, only the source areas can be used to analyze the spatial distribution pattern of landslides and to train the susceptibility model [72]. Therefore, on the basis of the above co-seismic landslide database, a landslide source area database was established based on Google Earth satellite images and UAV images to differentiate between the source area, circulation area and accumulation area of the landslide. Specifically, in cases where it was challenging to clearly identify the provenance area of a landslide, we selected the upper 50% of the landslide based on the elevation values and designated it as the provenance area [73]. Subsequently, mass points corresponding to the identified provenance areas were then extracted and employed as landslide sample points for further analysis. In addition, it is crucial to emphasize that all the landslide areas mentioned below specifically referring to the landslide source areas.).

M3. In Figure 8, the co-authors quoted the previous research results of the earthquake surface rupture zone, but the buffer zone did not follow the scope of the rupture zone. It should be adjusted. Although the SRL article (Xu et al., 2015) has been published, it is recommended that the co-authors mainly analyze it based on the seismic motion parameters, the re-located of the epicenter, and the topographical parameters. According to the reviewer’s research results (unpublished), this time The earthquake may be a deep rupture, the surface is shaken and deformed, and the non-fracture cracks to the surface. If it must be analyzed, the relationship between the dominant direction of aftershocks and the spatial distribution of earthquakes can be analyzed.

We sincerely apologized for the incorrect display of the buffer zone related to the distance to the seismogenic fault in Figure 8(g). The mistake was limited to the erroneous presentation of vector data regarding the seismogenic fault, but the buffer zones themselves were generated using accurate seismogenic fault vector data. We appreciate your advice, and as per your suggestion, we have re-drawn Figure 8(g) to rectify the error.

In addition to the SRL article (Xu et al., 2015), our previously published papers (Luo et al., 2020; Luo, 2020) conducted comprehensive analyses on the co-seismic surface rupture induced by the Ludian earthquake. Based on evidence from the particular distribution of aftershocks and co-seismic surface effects, it was established that the 2014 Ludian earthquake involved a complex faulting process associated with Riedel shear structures, i.e., NNW-striking sinistral strike-slip fault plane (Y shear) which trends approximately N40°W, and ENE-striking dextral strike-slip fault plane (R' shear) which is nearly perpendicular to former. Focal mechanism solutions were also derived using a generalized cut and paste method (Zhang et al., 2014) and by inverting the source rupture process (Liu et al., 2014; Xu et al., 2014). These solutions also indicate the involvement of a conjugate fault system, consisting of NNW-striking (~162°) and ENE-striking (~72°) faults, in the source rupture process of the 2014 Ludian earthquake.

Therefore, the selection of the seimogenic fault for the Ludian earthquake in this study is well-supported, making it suitable to as a factor for evaluating the susceptibility of landslides triggered by the Ludian earthquake.

Xu, X.W.; Xu, C.; Yu, G.H.; Wu, X.Y.; Li, X.; Zhang, J.G. Primary surface ruptures of the Ludian Mw 6.2 earthquake, southeastern Tibetan Plateau, China. Seismol. Res. Lett. 2015, 86, 1622–1635.

Luo, J.; Evans, S.G.; Pei, X.J.; Huang, R.Q.; Liu, M.; Dong, X.J. Anomalous co-seismic surface effects produced by the 2014 Mw 6.2 Ludian earthquake, Yunnan, China: An example of complex faulting related to Riedel shear structures. Eng. Geol. 2020, 266, 105476.

Luo, J. Slope dynamic response and formation mechanism of large-scale rockslide dam in the “8.3” Ludian earthquake. Chengdu University of Technology, 2020. (In Chinese)

Zhang, G.; Lei, J.; Liang, S.; Sun, C. Relocations and focal mechanism solutions of the 3 August 2014 Ludian, Yunnan M (s) 6.5 earthquake sequence. Chin. J. Geophys. 2014, 57 (9), 3018–3027 (in Chinese).

Liu, C.; Zheng, Y.; Xiong, X.; Fu, R.; Shan, B.; Diao, F. Rupture process of M (s) 6.5 Ludian earthquake constrained by regional broadband seismograms. Chin. J. Geophys. 2014, 57 (9), 3028–3037 (in Chinese).

Xu, L.; Zhang, X.; Yan, C.; Li, C. Analysis of the love waves for the source complexity of the Ludian M (s) 6. 5 earthquake. Chin. J. Geophys. 2014, 57 (9), 3006–3017 (in Chinese).

M4. Due to the relatively small-size magnitude of the Ludian earthquake and the small range of high-intensity areas, in terms of scale effects, the landform of the deep canyon at the epicenter determines the dominant direction of landslides, and the analysis results of earthquake cases are not universal. Meaning, once the landform structure changes, the so-called directionality will change.

The motivation for conducting this study originated from a curiosity about a specific phenomenon observed from the co-seismic landslides and the directional differences of seismological signals in the Ludian earthquake. Specifically, the largest landslide triggered by the Ludian earthquake was the Hongshiyan landslide, with a volume of approximately 12.24 Mm3. Interestingly, the opposite slope, which has a similar geologic structure but steeper topography, does not show obvious deformation associated with the earthquake. Moreover, the values of the PGA, PGV, Ia, and SED in the direction facing the Hongshiyan slope are significantly larger than those in the opposite slope. Therefore, we used a random forest model to conduct co-seismic landslide hazard assessment. The primary objectives of the study are as follows: (1) to elucidate the spatial distribution pattern of co-seismic landslides in the earthquake, an earthquake caused by a complex faulting process; (2) to determine the ground motion parameter that most significantly contributes to co-seismic landslides in the earthquake among PGA, PGV, Ia, and SED; (3) to examine the relationship between seismic energy variations in different directions and the corresponding varying degrees of slope instability.

Our analysis revealed that the distance to rivers and elevation were the primary factors influencing the spatial distribution of the Ludian earthquake-triggered landslides. The model's performance and its ability to accurately represent the spatial distribution of co-seismic landslides were essentially the same, regardless of whether the analysis incorporated PGA, PGV, Ia, or SED. Although the AUC of the model slightly decreases when the directional variation of ground motion parameters is taken into account, there is a notable reduction in the proportion of areas of ”high” and ”very high” landslide susceptibility. Therefore, it is suggested that the directional variation of ground motion parameters plays an essential role and should be taken into account in the co-seismic landslide susceptibility mapping for the Ludian earthquake. Moreover, in comparison to PGAd, PGVd and Iad, SEDd emerged as the most effective ground motion parameter for interpreting the distribution of the co-seismic landslides.

The research results largely align with our initial hypotheses. However, as you have pointed out, this study falls within the scope of a case study. Therefore, it would be valuable to conduct further analysis to enhance the universality and authority of the findings. This could involve addressing the limitations of this study, as discussed in the section of Disccusion, and exploring other earthquakes that exhibit similar phenomena, such as the 2018 Hokkaido earthquake and the 2016 kumamoto earthquake, among others.

M5. Other small problems:

a) The nature of the fracture in Line 149 Figure 1 should be marked in the legend, the first letter of the name of the fracture is capitalized, and consistent, the source of the intensity map, and the source of the fracture should be marked.

Based on your suggestions, we have made the following modifications: 1) we have updated the name of the fault in Figure 1 to reflect the revised information; 2) In the legend of Figure 1, we have included a label specifying the nature of the fault; 3) In the caption of Figure 1, we have stated the source of the intensity and fault maps (i.e., Intensity map published by the China Earthquake Administration. Source: http://www.gov.cn/xinwen/2014-08/07/content_2731360.html. The fault map was derived from Cheng et al. [51] and Luo et al. [52, 53].).

b) Line 249 Figure 6 emphasizes the statistical relationship between the landslide area and the slope where it is located. It has been pointed out earlier that the provenance area of the landslide is the most important statistical source, and the accumulation area of the landslide cannot also be included in the statistical scope.

Yes, the landslide areas mentioned in Section 2.3.4 “Slope aspect”, Table 3 and Figure 6 specifically refer to the provenance areas or source areas of the landslide. It is important to clarify that the statistics provided in these sections do not include the circulation areas and accumulation areas of the landslides. In addition, we have made the necessary revision by changing “landslide area” to “landslide source area” in the footnote of Table 3. Furthermore, we have introduced pertinent clarification that all the mentioned landslide areas specifically refer to the landslide source areas in Lines 206–207 (i.e., it is crucial to emphasize that all the landslide areas mentioned below specifically referring to the landslide source areas.).

c) It is suggested that the Discussion section should add the impact of landslide data integrity on the analysis results, especially after the distinction of landslide provenance areas.

The impact of completeness of the landslide inventory on the results of the analysis has been included in section 5.1 “Creating a comprehensive and detailed co-seismic landslide inventory is crucial for studying the spatial distribution of co-seismic landslides, assessing their susceptibility and understanding their impact on the geomorphic evolution of earthquake-affected areas. However, in this study, the resulting landslide inventory exhibits certain disparities in terms of the number of landslides due to the relatively limited coverage and the specific criteria used for interpreting landslides with areas exceeding 500 m2. It is important to note that despite these differences, the LAP obtained in our study (2.34%) is relatively close to the result of other more comprehensive inventories (2.71%). This similarity suggests that the landslide database we established possesses a certain level of representativeness and can be considered reliable for assessing co-seismic landslide susceptibility. Nonetheless, it is recommended to investigate the impact of inventory completeness, particularly in terms of the number of landslides, in subsequent research.).

The impact of distinguishing landslide source areas on the results has not been further discussed in the Discussion section. This is because the landslide sample sites used to analyze the spatial distribution pattern of landslides were specifically selected from landslide source areas. And this have been descripted in Section 2.2 (i.e. Since the primary focus of landslide susceptibility analysis revolves around the identification of potential landslide source areas, only the source areas can be used to analyze the spatial distribution pattern of landslides and to train the susceptibility model [72]. Therefore, on the basis of the above co-seismic landslide database, a landslide source area database was established based on Google Earth satellite images and UAV images to differentiate between the source area, circulation area and accumulation area of the landslide. Specifically, in cases where it was challenging to clearly identify the provenance area of a landslide, we selected the upper 50% of the landslide based on the elevation values and designated it as the provenance area [73]. Subsequently, mass points corresponding to the identified provenance areas were then extracted and employed as landslide sample points for further analysis. In addition, it is crucial to emphasize that all the landslide areas mentioned below specifically referring to the landslide source areas.).

Furthermore, the selection of landslide sampling areas and methods has been addressed in section 5.2 “Strategies for selecting landslide samples and prediction models” (i.e., Dou et al. [93] analyzed the accuracy of four sampling strategies (i.e., the centroid of landslide body, the landslide scarp centroid, samples of the landslide body, and samples of the scrap region) using logistic regression (LR), artificial neural network (NNET), and deep learning neural network (DNN) models, in order to perform the co-seismic landslide susceptibility mapping of the 2018 Hokkaido earthquake. Their results revealed that the most accurate strategy was landslide scarp, followed by landslide body, centroid of scarp, and centroid of body. However, the evaluation of the DNN model revealed that the accuracies of the four types of samples were largely similar, suggesting that the sampling strategy had little impact on the accuracy of DNN models with deep architecture. In contrast, Yi et al. [94] discovered that the accuracy of Convolutional Neural Network (CNN) models with the same deep architecture was influenced by the choice of sampling strategy at different scales, suggesting that the selection of samples and models remains a topic of ongoing debate within the field. The present study employed the centroid of scarp as the sampling strategy, which has been broadly used in previous studies. The evaluation model selected for the study is random forest, which is also a well-established and extensively used model in landslide susceptibility assessments. However, it is worth exploring different sampling strategies and deep learning models with complex architectures to gain a better understanding of the underlying causes of the unusual spatial distribution patterns of co-seismic landslides in the 2014 Ludian earthquake. Additionally, further analyses are required to investigate the possible reasons for the lower AUC of the model when the directional variation in ground motion parameter was taken into account.).

Round 2

Reviewer 2 Report

Thank you for the author's response and modifications.

Author Response

Thank you once again for your valuable time and expertise in reviewing this paper. Your insightful comments are greatly appreciated and have been instrumental in improving the manuscript.

Reviewer 4 Report

Thanks for your careful revision work, Here I added several minor points to improve the manuscript before accepted.

1.  The abstract section says the predominantly extent of coseismic landslides along major river systems with an NW-SE tread, I do not agree with this point, and suggest that it should be changed to maybe NE-SW tread, see Figs.3,4, there are two main regions, along the part of Niulanjiang River, and  along the Longtoushan town with NE-SW tread not NW-SE, As for the Hongshiyan site, the  biggest one which blocked the Niulanjiang River  is a rockfall along the right bank of the river with the slope aspect of about 180°(S), so I suggest co-authors should check this point and make a modify on this point both the Abstract and the Text;

2. Fig.2-Fig.8 should add several local names that can be identified  when the reviewers and readers talk about it; i.e. Longtoushan Town, Hongshiyan, Ganjiazhai;

3. To Fig.1, there is an earthquake with a magnitude of 5.0 near Zhaotong City missing the data, I suggest you add the data and the mainshock of Ludian with detail Year/Month/Day, so the readers can find where they, and move the compass at the right-top to another place in order to clear the location of another earthquake in the research area.

Author Response

We would like to thank you, most sincerely, for all the effort and expertise that you have contributed to reviewing. After two rounds of review, your insightful comments help us to improve the manuscript greatly. Especially, thank you for your valuable suggestion to including the main local names for the figures. We have carefully considered the comments and have revised the manuscript accordingly. We hope that the revised manuscript is acceptable. Detailed responses to your comments are given below.

M1. The abstract section says the predominantly extent of coseismic landslides along major river systems with an NW-SE tread, I do not agree with this point, and suggest that it should be changed to maybe NE-SW tread, see Figs.3,4, there are two main regions, along the part of Niulanjiang River, and along the Longtoushan town with NE-SW tread not NW-SE, As for the Hongshiyan site, the biggest one which blocked the Niulanjiang River is a rockfall along the right bank of the river with the slope aspect of about 180°(S), so I suggest co-authors should check this point and make a modify on this point both the Abstract and the Text;

We have revised “NW-SE trend” to “NE-SW trend” in Line 15, 295, and 621.

M2. Fig.2-Fig.8 should add several local names that can be identified when the reviewers and readers talk about it; i.e. Longtoushan Town, Hongshiyan, Ganjiazhai;

Following your suggestions, we have included the main local names in Figure 2, 3, 4, 5, 7, 8, and 11.

M3. To Fig.1, there is an earthquake with a magnitude of 5.0 near Zhaotong City missing the data, I suggest you add the data and the mainshock of Ludian with detail Year/Month/Day, so the readers can find where they, and move the compass at the right-top to another place in order to clear the location of another earthquake in the research area.

We sincerely apologize for the oversight of not including the time information for two historical earthquakes in Figure 1. In the revised version, we have rectified this mistake and included the time information for these two historical earthquakes. Furthermore, we have provided detailed information about the 2014 Ludian earthquake and made the necessary adjustment by relocating the compass to the top-left position. Thank you for bringing this to our attention, and we appreciate your understanding.
